# 🎇 FLARE-AI: Flaw Reporting for AI

**Shayne Longpre** [⋆ 1]   **Elaine Zhu** [⋆ 2]   **Carson Ezell** [⋆ 3 4]   **Avijit Ghosh** [⋆ 5]

**Sean McGregor** [⋄ 6]   **Kevin Paeth** [⋄ 7]   **Kevin Klyman** [⋄ 8 3]   **Sayash Kapoor** [⋄ 9]   **Rishi Bommasani** [⋄ 8]   **Ruth Appel** [⋄ 8]

**Gregory Strom** [10]   **Lauren McIlvenny** [10]   **Mark M. Jaycox** [11]   **Peter Slattery** [12]   **Nathan Butters** [13]

**Arvind Narayanan** [† 14]   **Percy Liang** [† 8]   **Alex Pentland** [† 1]

\* Lead authors ⋄ Top contributors † Advisors

## Abstract

Flaw reporting for deployed AI systems is fundamental to identifying system failures and improving AI safety. Yet the AI reporting ecosystem is fragmented: researchers who identify flaws often do not know what or where to report, and groups who receive reports rarely share them with other relevant stakeholders. As a result, good-faith reporters duplicate effort by submitting many different forms, and recipients lack standardized, triage-ready information. We survey 12 reporting systems published by AI developers, cybersecurity groups, and AI flaw aggregators, identifying five recurring design challenges spanning discoverability, scope, information collection, coordination, and guidance for strict-liability cases. Building on this analysis and feedback from 49 experts across 32 organizations representing developers, security researchers, and ecosystem coordinators, we introduce FLARE-AI, an open-source AI flaw reporting system designed for interoperability with existing systems. FLARE-AI streamlines flaw report creation by collecting triage-relevant information through conditional logic and early classification, then enables optional dissemination of standardized, machine-readable reports to multiple developers, coordinators, and incident registries from a single submission. By lowering barriers to reporting AI flaws and improving interoperability across stakeholders, FLARE-AI helps break down silos and accelerate remediation across the AI ecosystem.

## 1. Introduction

Independent, third-party evaluation is a cornerstone of AI accountability (Raji et al., 2022; McGregor et al., 2024a; Householder et al., 2024; Stosz et al., 2025). While static benchmarks enable standardized comparisons, many high-impact risks are discovered through dynamic evaluation and real-world observation: safety and security testing, red teaming, and the investigation of deployed system failures (Angwin et al., 2016; Raji & Buolamwini, 2019; Zou et al., 2023; Carlini et al., 2024; Hughes et al., 2024). To remediate flaws, researchers must be able to report them to the right stakeholders, and recipients must be able to triage and respond (McGregor, 2021; Cattell et al., 2024b).

Today, AI flaw and incident reporting is neither streamlined nor interoperable (Longpre et al., 2025). Reporting channels are difficult to discover and onerous to complete, reporters must decipher what flaws are in scope for each channel, and when flaws implicate multiple entities, reporters must submit duplicative reports with little guidance on which stakeholders should receive them. Compounding this burden, the ecosystem remains siloed: recipients rarely coordinate with other affected parties, delaying mitigation when flaws affect multiple systems. Emerging governance frameworks such as the EU AI Act (Article 73) (European Parliament, 2024) and the U.S. AI Action Plan (The White House, 2025) are beginning to mandate incident reporting, underscoring the need for standardized, interoperable infrastructure that enables consensus on what constitutes a reportable incident and how to respond effectively.

---

[1] Massachusetts Institute of Technology [2] Northeastern University [3] Harvard University [4] RAND Corporation [5] Hugging Face [6] AVERI [7] UL Research Institutes [8] Stanford University [9] Mozilla [10] CERT, Carnegie Mellon University Software Engineering Institute [11] Google [12] MIT FutureTech [13] AI Risk and Vulnerability Alliance [14] Princeton University. Correspondence to: Shayne Longpre <shayne.r.longpre@gmail.com>, Avijit Ghosh <avijit@huggingface.co>.

*Proceedings of the $43^{rd}$ International Conference on Machine Learning*, Seoul, South Korea. PMLR 306, 2026. Copyright 2026 by the author(s).

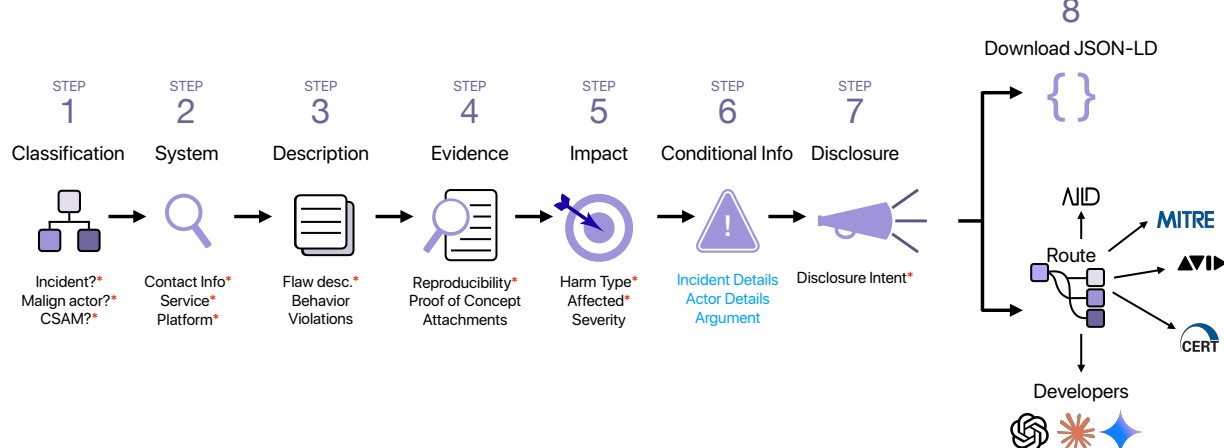

*Figure 1.* **FLARE-AI Reporting Workflow Overview.** The system guides reporters through eight steps from initial classification to report generation and routing. Fields marked with asterisks (*) are required. Blue text indicates conditionally displayed fields based on the Step 1 classification questions. The routing partners in the image represent real stakeholder commitments at the time of writing.

To design a practical reporting system for this setting, we combine ecosystem analysis with stakeholder input. We conduct a detailed survey of 12 prominent AI reporting systems, comparing their in-scope definitions, taxonomies, information collection, and post-submission coordination practices, and collect feedback from 49 experts across 32 organizations representing AI developers, flaw coordinators, security researchers, and policy organizations. From these efforts, we identify five recurring design challenges: (1) discoverability and process transparency, (2) ambiguous in-scope coverage and incompatible taxonomies, (3) inconsistent information collection and missing triage details, (4) limited interoperability, sharing, and coordination, and (5) insufficient guidance for strict-liability cases.

We address these challenges with FLARE-AI, an open-source AI flaw reporting system developed in collaboration with key stakeholders including coordination bodies (CERT, MITRE), incident databases (AIID), infrastructure providers (Hugging Face), international organizations (OECD), and model developers (OpenAI, Anthropic, Google). FLARE-AI broadens intake (all flaws, hazards, vulnerabilities, and incidents are in scope) while structuring triage through early classification and conditional routing. FLARE-AI enables reporters to generate and download reports themselves, without any storage or routing, in case they wish to handle dissemination entirely themselves. Most importantly, FLARE-AI enables optional dissemination of standardized, machine-interpretable reports to multiple developers, coordinators, and registries from a single submission, reducing duplicated effort and improving interoperability across the reporting ecosystem. An early, anonymized demo is available here: `https://ai-flaw-reporting-frontend-app.vercel.app/`.

Our main contributions span ecosystem analysis, system deisgn, and infrastructure construction – each grounded in empirical study of existing systems and iterative consultation with the stakeholders who will receive and triage these reports:

1. **The first extensive survey of AI reporting systems:** We analyze 12 prominent AI flaw and incident reporting schemes, characterizing their scope, taxonomies, information requirements, and coordination practices, and distill five ecosystem-wide design challenges.

2. **FLARE-AI, a novel AI flaw reporting system:** We introduce a reporting system that balances ease of reporting with triage-relevant detail using early classification, conditional logic, and few required questions.

3. **Interoperable, one-stop dissemination across groups:** In collaboration with infrastructure providers, cybersecurity centers of excellence, and model developers, including CERT, MITRE, and Hugging Face, we enable standardized, machine-readable report generation and optional dissemination to multiple stakeholders, reducing silos and duplicative reporting.

## 2. Background and Problem Statement

Billions of people use general-purpose AI (GPAI) systems, but infrastructure for identifying, reporting, and remediating flaws that users find has not kept pace with AI deployment (Bengio et al., 2025). Developers and deployers of GPAI cannot identify all of the critical flaws in their systems, particularly when systems are deployed in contexts not contemplated during development. Sourcing flaw reports from expert external researchers as well as users can provide

*Table 1.* Definitions of Key Terms

| Term | Definition |
| --- | --- |
| Flaw | A set of conditions or behaviors that allow the violation of an explicit or implicit policy related to the safety, security, or other undesirable effects from use of the AI system (Cattell et al., 2024a; Walshe & Simpson, 2022; Longpre et al., 2025). |
| Incident | A real-world event that has resulted in harm, loss, or policy violations (OECD, 2024; McGregor, 2021; Dixon & Frase, 2024a). |
| Hazard | A set of conditions that may lead to an incident; commonly used by safety engineers (Khlaaf, 2023; OECD, 2024). |
| Vulnerability | A set of conditions that may lead to an incident; commonly used by security professionals for software security threats (Leveson, 2019; Householder et al., 2017b; Anderson et al., 2023). |

a greater diversity of subject matter knowledge, novel approaches to flaw identification, independence, and greater speed and throughput.

Throughout this work, we use the terms flaw, incident, hazard, and vulnerability as defined in Table 1. *Flaw* is the broadest category, encompassing all issues that may or may not involve malicious intent and may or may not have already caused harm.

To enable such external reporting, a number of developers and cybersecurity groups have established channels for reporting AI flaws, ranging from highly structured forms to open-ended email inboxes. Yet the current reporting ecosystem for AI flaws has three fundamental issues:

1. **Reporting channels are not discoverable:** It is difficult for even experienced AI researchers to understand where they should submit reports of AI flaws. Developers often have two or three separate emails for different kind of reporting, causing confusion. This contributes to a poor reporting culture, where many jailbreaks are posted on social media; sometimes irresponsibly.

2. **Reporting channels are not interoperable:** Forms from different organizations are not interoperable in that they collect different information, use different taxonomies of harm and scopes of what is reportable, and operate under different terms regarding what can be shared publicly or with other organizations. As a result, reporting a flaw to one developer does not necessarily speed the process of reporting it to another.

3. **Reporters bear the burden of dissemination:** Recipients of AI flaw reports do not by default share those reports with other groups. This poses an acute issue for

universal flaws, which can affect many GPAI systems simultaneously (Zou et al., 2023). For each impacted stakeholder to receive notice, the individual who discovers the flaw must report to each of them.

Our work builds on established practices in coordinated vulnerability disclosure, bug bounty programs, and AI evaluation methodologies. A comprehensive review of related work on vulnerability disclosure frameworks, AI red teaming, incident databases, and transparency mechanisms is provided in Appendix A.

To understand these problems in depth and design effective solutions, we conducted a comprehensive study of the existing reporting ecosystem and engaged with key stakeholders across the AI safety and security community.

## 3. Methodology

The many AI flaw reporting systems offer competing points of entry to flaw reporting. There is limited interoperability or sharing of these flaws, especially between AI developers, despite clear applicability between them, and security/safety benefits to sharing this information. In this work, we assess flaw reporting systems to build an open source reporting form which connects the various flaw networks, rather than competing with, or replacing them. This requires ensuring the collected information bridges the various entry point requirements, and remains suitably easy both to submit reports and to triage them. To design an AI flaw reporting system, we (a) reviewed 12 existing reporting forms and schema proposals, and (b) collected feedback from experts across various organizations representing three core groups: AI flaw report recipients (including AI developers, flaw coordinators), AI flaw reporters (AI and security researchers), as well as other general experts in AI flaw reporting.

### 3.1. Comparative Analysis of Existing Reporting Systems

Among the analyzed forms, we selected 12 of the most prominently discussed, from among those currently offered by popular AI developers (OpenAI, Anthropic), AI incident reporting databases (AVID, AIID (McGregor, 2021)), security vulnerability coordination organizations with a focus on AI (CERT, CISA, MITRE ATLAS (Liaghati, 2025)), and other organizations now collecting or proposing collection of AI-related flaws (AIAAIC, Mozilla's 0DIN, OECD AI Incidents and Hazards Monitor (AIM) (OECD, 2024)). Note that some organizations like CISA have more reporting forms, but we chose to select a representative subset spanning organizations.

We analyzed these forms across two dimensions. First, we examined high-level characteristics including scope (what

types of issues are reportable), taxonomies used for classification, reportable systems (vendor-specific or universal), number of required and total fields, presence of CSAM guidance, sharing and coordination policies, and public disclosure pathways. Second, we conducted detailed field-level analysis examining what specific information each form collects, such as reporter information, system identification, flaw description, impact assessments, affected stakeholders, reproducibility steps, and evidence requirements. The full comparison methodology and detailed analyses are provided in Appendix B.

### 3.2. Multi-Stakeholder Consultation Process

To refine our understanding of design challenges and validate potential solutions, we conducted consultations with 49 experts across 32 organizations between June and August 2025 (Table 2). Stakeholders represented AI developers, AI safety organizations, vulnerability coordination bodies, incident databases, academic researchers, bug bounty platforms, and policy organizations. They provided feedback through demonstrations of early prototypes, written responses to structured questions, and iterative design discussions. For red teamers and security researchers, sessions were additionally structured as user studies in which participants filed reports against real flaws they had previously identified, with their reporter experience captured as feedback (D.1). Table 6 documents specific design changes made in response to this consultation process.

*Table 2.* Stakeholders Consulted in FLARE-AI Development

| Stakeholder Type | Organizations | Count |
|---|---|---|
| AI Developers | Hugging Face (1), Anthropic (2), OpenAI (2), Google (1), Cohere (1), Meta (1), NVIDIA (1) | 9 |
| AI Safety Orgs. | Humane Intelligence (2), Thorn (1), AI Village (1), FMF (1) | 5 |
| Coordination Bodies | CERT CC (2), CISA (1), MITRE (1) | 4 |
| Incident Databases | ARVA/AVID (3), AIID (1), MIT AI Risk Repository (2) | 6 |
| Academic Researchers | MIT (3), Stanford (5), Berkeley (2), Princeton (2), Northeastern (1), Harvard (1) | 14 |
| Bug Bounty Platforms | Bugcrowd (1), HackerOne (2) | 3 |
| Other Organizations | OECD (2), Partnership on AI (1), ML Commons (1), Center for AI and Digital Policy (1), UL Research (1), RAND (1), IBM Research (1) | 8 |
| **Total** | **32 organizations** | **49** |

The consultation revealed several critical tensions in form design. Reporters prioritized simplicity and minimal submission barriers, while recipients emphasized comprehensive information for effective triage. Child safety experts highlighted risks of inadvertent illegal material distribution, particularly for CSAM. Security professionals stressed compatibility with existing frameworks like CVE and CWE. Academic researchers identified taxonomic ambiguities and noted that harm severity assessments must distinguish individual-level from societal-level impacts. These tensions informed the design decisions we describe in Section 5.

The complete consultation methodology, detailed feedback by stakeholder type, and specific design refinements are documented in Appendix D.

## 4. Survey Results: Five Design Challenges

In this section, we present the results of our comparative survey of 12 AI flaw and incident reporting systems. Through our analysis and stakeholder consultations, we identify five recurring design challenges that hinder effective flaw reporting in the current AI ecosystem. These challenges informed the design of FLARE, which we present in Section 5.

All reporting systems were compared across fine-grained criteria in Figure 3 and overall design in Figure 2. Overall, we find that **AI reporting options are inconsistent with one another and often silo their data**.

This fragmentation creates two compounding problems: reporters struggle to identify the right venue(s) and must often duplicate effort across systems, while recipients frequently lack standardized, triage-relevant information and mechanisms for coordination across stakeholders. We organize these observations into five design challenges: (1) discoverability and process transparency, (2) interpreting in-scope coverage and taxonomies, (3) balancing reporter burden with triage-relevant information, (4) interoperability, sharing, and coordination, and (5) guidance for strict-liability cases. We detail each challenge below.

### 4.1. Discoverability and Process Transparency

In discussions with AI flaw researchers, we found many were unaware of existing reporting options–particularly OpenAI and Anthropic's pathways, or which applied to specific flaws. Awareness of vulnerability coordinators like CERT, CISA, or MITRE was even sparser among AI researchers than software security experts.

Researchers flagged limited clarity on core process details: anonymity, public disclosure, and downstream coordination. Our survey (Figure 2) reveals meaningful differences: only 3 of 12 allow anonymity, 9 allow public disclosure, and 8 conduct cross-organization coordination. Without clearer guidance or standardization, researchers may hesitate to adopt these tools.

### 4.2. Ambiguous In-Scope Coverage and Incompatible Taxonomies

In Figure 2, we find substantial variation in what systems accept as in-scope. In the case of OpenAI, multiple reporting options may cover the same category. Even where systems clarify scope, like Google's Vulnerability Reporting Program (VRP), limitations apply: Google defines four prompt attack categories but only three are in-scope for its VRP,

| | Anthropic (HackerOne) | Google (AI VRP) | OpenAI (Report Content) | OpenAI (Bugcrowd) | CERT CC | CISA | MITRE ATLAS | AVID | Mozilla | AIID | AIAAIC | OECD | FLARE-AI |
|---|---|---|---|---|---|---|---|---|---|---|---|---|---|
| **Form Exists?** | ✓ | ✓ | ✓ | ✓ | ✓ | ✓ | ✓ | ✓ | ✓ | ✓ | ✓ | ✓ | ✓ |
| **Allows Reporter Anonymity?** | ✓ | | | | ✓ | | | | | ✓ | | | ✓ |
| **Reportable Systems** | Anthropic Systems | Google Systems | OpenAI Systems | OpenAI Systems | Any | Any | Any | Any | Any | Any | Any | Any | Any |
| **Scope** | Vulnerability | Vulnerability | Safety Incident | Vulnerability | Vulnerability | Security Incident | All Incidents | Vulnerability, Hazard | Vulnerability, Hazard, All Incidents (prev. reported only) | Hazard, Vulnerability, All Incidents | Hazard, Vulnerability, All Incidents | Hazard, Vulnerability, All Incidents | Hazard, Vulnerability, All Incidents |
| **Taxonomies** | Flaws, Tactics | Flaws | Harms | Flaws | None | Impacts | Impacts, Harms | Domains | Tactics | None | None | Harms | Flaws, Harms, Tactics, Impacts |
| **Required Fields** | 5 | 8-9 | 6-9 | 4 | 13 | 18-20 | 15 | 6 | 9 | 7 | 9 | 7 | 6 |
| **Total Fields** | 9-20 | 10-12 | 12-15 | 7 | 23-29 | 28-45 | 53 | 13-19 | 10 | 19 | 10 | 29 | 17-30 |
| **Info Covered** | 10 | 12 | 8 | 11 | 13 | 17 | 19 | 11 | 9 | 9 | 9 | 21 | 22 |
| **CSAM Guidance?** | | | | | | | | | | ✓ | | | ✓ |
| **Sharing & Coordination** | ✓ | ✓ | | ✓ | ✓ | ✓ | ✓ | | ✓ | | | ✓ | ✓ |
| **Public Disclosure** | ✓ | ✓ | | ✓ | ✓ | | | ✓ | ✓ | ✓ | ✓ | | |

*Figure 2.* **A high-level comparison of AI flaw/incident reporting design.** Table 5 defines the specific criteria we used to evaluate the scope of flaws, taxonomies, reportable systems, rigor of information collected, and other coordination choices. Table 3 details the reporting forms.

with content violations directed to separate channels. Systems often restrict to specific categories (incidents only, vulnerabilities only) or accept multiple types. Determining scope requires careful documentation review yet often remains ambiguous.

Additionally, the taxonomies used to capture the type of flaw, its impact, and potential harms vary widely. For instance, the AI Incident Database (AIID) referenced an existing harm taxonomy (Hoffmann & Frase, 2023), whereas AIAAIC and OECD developed their own harm taxonomies (Abercrombie et al., 2024; OECD, 2023). Multiple academic papers and organizations have attempted to catalog AI flaws and impacts to provide a shared vocabulary for rapid triage (MITRE Corporation, 2024; Klyman, 2024; Zeng et al., 2025; Weidinger et al., 2022). However, standardization and interoperability across taxonomies is largely absent, meaning forms collect information that is not easily translated across systems.

### 4.3. Inconsistent Information Collection and Missing Triage Details

On multiple occasions, AI researchers approached us with flaws that implicated multiple AI developers. Their primary question was: where should they report the flaw to follow responsible disclosure practices? The answer was often: multiple reporting systems, each with distinct information requirements, expectations, and functionality—an onerous burden for reporters.

The information collected across systems, *even when they cover the same or similar scope,* differs substantially–from extensive (53 fields for MITRE ATLAS) to sparse (7 for OpenAI's Bugcrowd form). Collecting more information enables faster triage, but less information reduces reporter burden and increases submission likelihood. A second dimension is which fields are optional versus required. The central challenges are (a) lack of standardization limiting interoperability, and (b) mismatch between reporter burden and recipient needs.

For example, developers emphasized that models are offered through many products and services. Claude, for instance, spans model families, versions, modes (e.g., extended thinking), and access pathways (website, API, or embedded products like Cursor). To remediate effectively, developers need the model, version, access channel, and location (relevant for local laws). However, most systems collect only the model name, or additional details via open ended, non-standardized fields (such as Google VRP), thereby limiting intake of actionable information that would speed up triage.

### 4.4. Limited Interoperability, Sharing, and Coordination

Our survey also revealed substantial differences in how reports are handled after submission. There is little evidence that AI developers make reports public or share them with competitors, even when models could plausibly share the

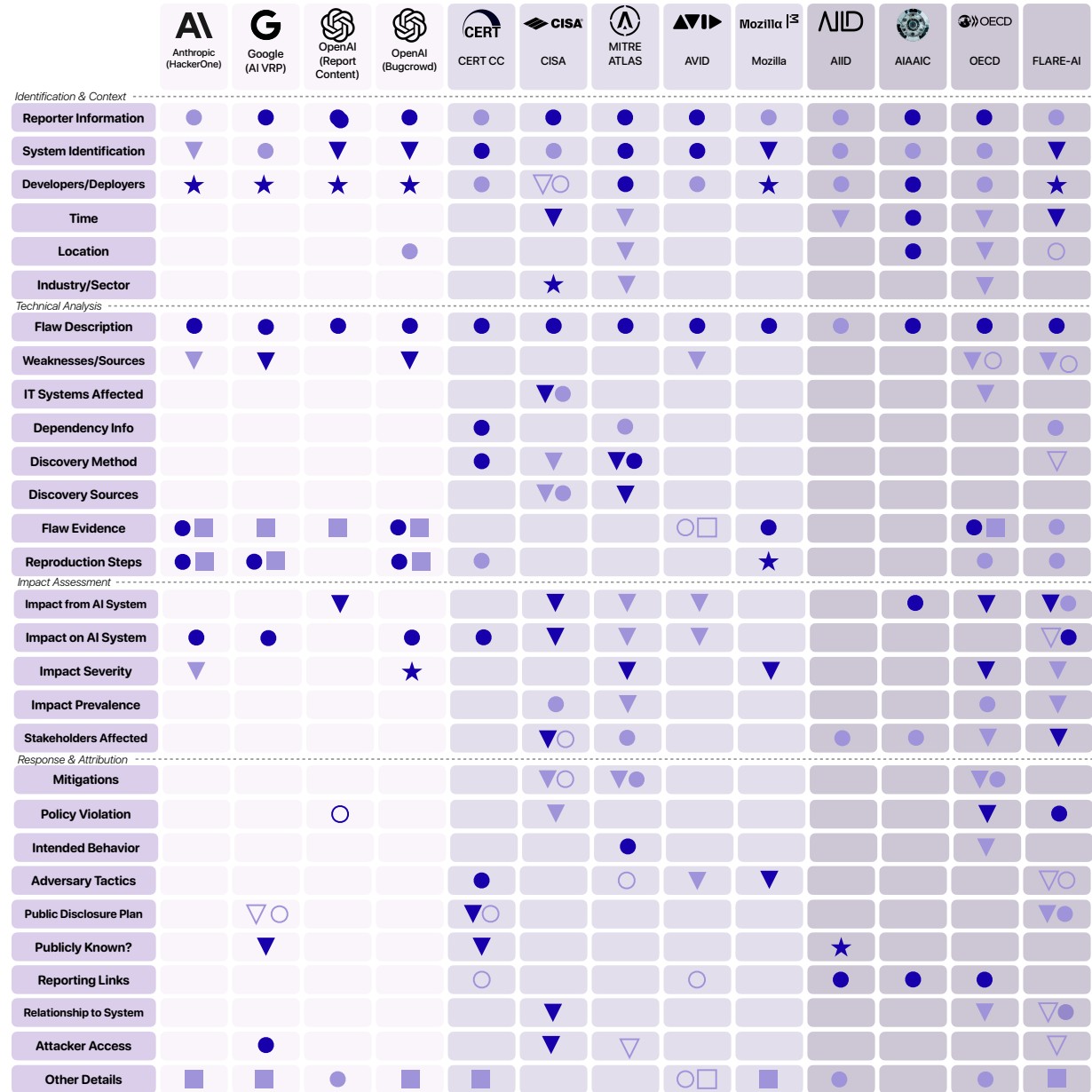

*Figure 3.* **A detailed comparison of AI flaw/incident reporting design.** Table 3 details the reporting forms. Table 4 defines the specific criteria we used to evaluate the presence or absence for each element of a form. **Key:** Dark shades are required, light shades are optional, unfilled shapes are conditional; circle (e.g. ●) is text entry, square (e.g. ■) is document upload, triangle (e.g. ▼) is dropdown, ★ is inferred, empty is not collected.

same vulnerabilities. And when coordinating organizations do share reports, there does not appear to be a machine-readable standard that supports broader interoperability and analysis of this highly heterogeneous data. We believe this is important, because researchers indicated they often needed to complete many distinct forms with largely overlapping information to inform all relevant stakeholders. This bur-

den likely dampens reporting and leaves systems vulnerable when reporters submit to only one developer, and that developer does not share findings due to competitive incentives.

Moreover, while organizations like CERT, CISA, and MITRE offer powerful coordination infrastructure for software vulnerabilities and have begun integrating AI-related vulnerabilities into their workflows (Householder et al.,

2024), the AI flaw reporting ecosystem has yet to fully leverage these established coordination mechanisms. Greater integration with these bodies represents an important opportunity to strengthen AI-specific issue coordination.

Flaw recipients and coordination organizations also emphasized that public disclosure information (whether the reporter has already disclosed the issue publicly, or intends to do so imminently) is critical for triage and prioritization, yet is not collected by any form except CERT's and Google's VRP (see Figure 3).

### 4.5. Insufficient Guidance for Strict-Liability Cases

Certain flaw or incident reports may involve strict liability offenses, such as the generation of real or synthetic Child Sexual Abuse Material (CSAM) (Thorn, 2024). Image and video generation models are particularly susceptible to these flaws, and these failures are already causing significant harm worldwide (Hawkins et al., 2025; Kamachee et al., 2025). In such cases, even possession (or submission) of documentary evidence may be illegal. However, few of the reporting systems provide checks or guidance for these types of reports—a critical gap in existing infrastructure. In one instance, Google's Vulnerability Reporting Program directs researchers to its Legal Content Reporting flow, which provides the public (and researchers) the ability to report illegal content.

## 5. FLARE-AI: Our AI Flaw Reporting System

Having identified five key design challenges in existing AI flaw reporting systems, we now present FLARE-AI, an open-source reporting system that systematically addresses each challenge. FLARE-AI was developed through iterative collaboration with downstream stakeholders including CERT, MITRE, AIID, Hugging Face, OECD, and several AI developers (OpenAI, Anthropic, Google). Several of these organizations provided direct input that shaped specific design decisions (Table 6) and have made commitments to integrate with the routing layer.

A demo of FLARE-AI is available here: `https://ai-reports.org`. Figure 1 shows the system's workflow.

### 5.1. Addressing Discoverability Through Centralized Information

In Subsection 4.1 we found that AI researchers were often unaware of reporting options, unsure which were relevant, and unclear on key differences between systems. Accordingly, the "Information & Resources" page compiles 15 reporting systems spanning AI developers, flaw/incident databases, and flaw coordination agencies. Each entry explains when to report, and the list is sortable by organization type and by in-scope flaw types. This provides a simple starting point for reporters to identify the most relevant venues for responsible disclosure.

### 5.2. Addressing Scope and Taxonomy Challenges Through Broad Intake

In Subsection 4.2 we found substantial variation in what systems accept as in-scope and in the taxonomies they use to classify flaws, impacts, and harms. Informed by prior work (Longpre et al., 2025), we therefore do not restrict the definition of flaws: all flaws, hazards, vulnerabilities, and incidents are in scope. This broader scope increases coverage and reduces false negatives at intake, but requires more structured collection to distinguish hazards from incidents, and vulnerabilities from broader AI safety issues. We address this with three Yes/No questions in the first stage of the form:

1. **An *Incident*:** Has this flaw already caused an incident of harm (e.g., harm to people, rights, property, or infrastructure)?

2. ***Malicious Actor*:** Could this flaw be used by someone with bad intent?

3. ***Strict Liability*:** Does this involve child sexual abuse material (real or synthetic)?

The first two questions classify the issue and determine downstream form structure, routing reports to appropriate pathways (incidents, vulnerabilities, hazards, or security concerns). The third question serves as a critical safety check: CSAM reports trigger specialized guidance to appropriate authorities and do not proceed through standard submission.

For flaw, impact, and harm taxonomies, we selected general-purpose options that are widely recognized, frequently updated, and likely to remain interoperable as standards evolve. Our harm taxonomy draws on that from AIAAIC, which was designed to cater to broad audiences, remain extensible, and be interoperable with other taxonomies (Abercrombie et al., 2024). Our flaw taxonomy draws on OWASP Top 10 lists of AI-related issues, which are widely recognized and regularly updated (OWASP, 2024). Because we open-source the code, our taxonomies can also evolve to meet stakeholder needs. Downstream adopters can introduce additional failure modes into existing taxonomies, or swap in alternative taxonomies as needed.

### 5.3. Addressing Information Collection Through Conditional Logic and Routing

In Subsection 4.3 we found that systems collect highly variable amounts of information and often omit key triage details. This creates a recurring tension: minimizing reporter burden versus collecting sufficient information for repro-

duction, prioritization, and mitigation. We mitigate this trade-off with three mechanisms. First, we use the Stage 1 classification to conditionally route users, asking only questions relevant to their case (e.g., malicious-actor questions only when malicious use is plausible). Second, we pair an optional "long" path (up to 30 input fields) with a small required core (6), enabling both lightweight and rigorous submissions. Third, we reuse earlier inputs to streamline later steps: based on selected products/systems, we surface relevant policy links under "Potential Policy Violation," and we suggest plausible reporting destinations based on the report content. Together, these choices reduce friction while improving triage-relevant specificity for recipients.

### 5.4. Addressing Coordination Through Automatic Dissemination

In Subsection 4.4 we found that flaws often implicate multiple AI systems, yet reports are rarely shared beyond the initial recipient. Even when coordinating organizations receive reports, they may not reach all affected developers. As a result, reporting is siloed and the burden of notifying stakeholders falls on reporters, who must duplicate effort across many forms.

A primary innovation of FLARE-AI is automatic dissemination of interoperable reports to multiple stakeholders, enabled through coordination with downstream recipients. FLARE-AI is stateless by default: reporters can generate and download reports locally without server-side storage or identity requirements, then optionally disseminate to selected recipients. If they choose, they can then select from dozens of AI developers, security agencies, flaw databases, and coordinators to automatically send the report via a mix of APIs and official email channels. This keeps control with the reporter while removing the need to search for and complete many distinct reporting forms.

To enable this dissemination, FLARE-AI produces machine-readable and machine-interpretable reports structured as JSON-LD, enabling automated processing, routing, and integration into existing vulnerability management workflows. The machine-readable format enables automated routing: when a report tags a specific model as the affected system, our dissemination logic automatically includes the relevant developers, security teams, coordinators, and incident databases in the distribution list, ensuring that reported flaws contribute to broader pattern analysis that can help other developers identify similar risks proactively. Our reports are designed for compatibility with established systems including CVE/CWE (for traditional security vulnerabilities), AVID (for AI-specific flaws), CERT Coordination Center (for multi-party vulnerability coordination), and CISA (for government and critical infrastructure coordination). Full technical specifications for the machine-readable report for-

mat and integration details are provided in Appendix C.4.

### 5.5. Addressing Strict-Liability Cases Through Specialized Guidance

In Subsection 4.5 we found that most reporting systems provide little guidance for strict-liability cases, such as real or synthetic child sexual abuse material (CSAM), for which even possession can be illegal. We therefore ask about CSAM in Stage 1 and provide guidance on where users should report such material, rather than submitting it through our system. While not a singular differentiator, we believe this guidance is essential given the severity of harm and the prevalence of these incidents (Thiel et al., 2023; Internet Watch Foundation, 2024).

### 5.6. Adaptability and Open Source

GPAI systems, the services they are embedded in, and the standards for reporting flaws are in flux, with multiple standards development organizations (ETSI, ISO) actively developing such standards (International Organization for Standardization & International Electrotechnical Commission, n.d.; Cybersecurity and Infrastructure Security Agency, 2025). As an open-source and modular system, FLARE-AI can be updated as the needs of both evaluators and organizations evolve. Reporting steps can be updated independently to collect additional types of information, generated reports can adopt new preferred taxonomies, and new coordination recipients can be easily added. This flexibility is essential as technical standards and community norms mature.

### 5.7. Initial Empirical Signals

Although our primary contribution is the design and construction of FLARE-AI, we have begun collecting evidence on whether the system meets its goal in practice. The expert assessments in Table 2 included user-study-style sessions with security researchers, in which participants used early prototypes to file reports against real flaws they had previously identified, with their experience captured through structured feedback (Appendix D.1). Participants identified specific friction points, such as terminology ambiguities, missing "unknown" options, file-upload context fields, etc., that we resolved iteratively (Table 6).

## 6. Discussion

Flaw reporting for AI is not working at present. GPAI systems present more risks than ever, yet reports are either not made, do not contain critical information, or do not reach the relevant parties. Our survey of existing flaw reporting systems shows that reporting mechanisms are hard to discover, ambiguous in scope, inconsistent in their data collection, inadequate in their guidance for strict liability cases, and

not interoperable. Compared to vulnerability reporting for software systems, flaw reporting for AI is decades behind.

Reporting infrastructure like FLARE-AI helps tackle this pressing issue. By making reporting easier, broader, and more triage-relevant, tools like ours can increase the speed, scale, and mitigation potential of AI flaw reporting. Through FLARE-AI, we build this infrastructure to address this challenge and have coordinated with key players across the ecosystem to promote adoption and alignment.

**Limitations and Future Work**    While we engaged with a variety of experts and organizations, our stakeholder consultation focused primarily on North American and European organizations, though the frameworks we drew upon (including those approved by 38 OECD countries and 44 nations across Asia, Latin America, and Oceania) provide some international validation. Future work should incorporate perspectives from underrepresented regions to ensure global applicability.

A fundamental challenge for any unified reporting system is balancing comprehensiveness with usability. Each existing framework emphasizes different aspects of incident reporting at varying levels of granularity. We address this through conditional logic and progressive disclosure: maintaining minimal required fields while enabling detailed optional reporting, and machine-readable formats that allow recipient-specific transformations. However, this may not fully eliminate the tension between universal accessibility and framework-specific rigor; some recipients may require information beyond what any single form can reasonably collect, necessitating follow-up communication.

Future research directions include empirical evaluation of FLARE-AI's impact on reporting rates, triage efficiency, and coordination effectiveness compared to existing systems. Additionally, extending FLARE-AI to support AI supply chain vulnerabilities, where flaws propagate through model fine-tuning, API dependencies, or shared training data, represents an important frontier for future development.

## Impact Statement

This work aims to strengthen AI safety and accountability infrastructure by reducing barriers to responsible flaw disclosure and improving coordination across stakeholders. By making it easier for researchers, users, and security experts to report AI system failures, FLARE-AI can accelerate the identification and remediation of risks that affect millions of people worldwide. The standardized reporting framework may help prevent repeated harms by ensuring affected parties receive timely notification of issues, particularly for vulnerabilities that span multiple AI systems.

**Policy Implications**    As regulatory frameworks such as the EU AI Act mandate incident reporting for high-risk AI systems (European Parliament, 2024), disclosure infrastructure like FLARE-AI can inform regulatory approaches to information sharing across jurisdictions while respecting different legal regimes. Policymakers should consider how disclosure timelines balance security research incentives against harm prevention, and how to support equitable access to reporting infrastructure globally. Safe harbor protections for good-faith security researchers remain essential, and centralized coordination bodies require governance mechanisms that maintain stakeholder trust across competing interests.

We emphasize that FLARE-AI is currently positioned as an *ecosystem coordination* tool rather than a *compliance reporting* tool: it does not directly format submissions for specific regulators such as the EU AI Office or other national authorities. However, the schema is extensible by design and several organizations FLARE-AI routes to maintain existing channels and triage infrastructure for forwarding reports to the relevant authorities.

**Risks and Mitigation**    However, this infrastructure also introduces risks warranting careful consideration. Centralizing flaw information creates potential targets for attackers seeking to exploit unpatched vulnerabilities, and dissemination mechanisms could be abused without proper access controls. The framework's emphasis on interoperability may inadvertently advantage larger organizations with resources to integrate machine-readable reports, while smaller developers face higher adoption barriers. Defining what constitutes a reportable flaw involves normative choices about acceptable AI behavior, and our taxonomies necessarily reflect particular perspectives on risk that other stakeholders may dispute. While we provide guidance on handling illegal content, the existence of reporting infrastructure does not eliminate difficult questions about researcher liability and safe harbor protections.

Our system's voluntary adoption model and stateless design mitigate some risks: reporters maintain full control over dissemination, and FLARE-AI does not centrally store reports. However, responsible deployment requires ongoing attention to misuse potential, equity of access, and the evolving landscape of AI system failures. We believe the benefits of improved coordination outweigh these risks when implemented with appropriate safeguards.

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

# Appendix

## A. Related Work

### A.1. Foundations of Coordinated Vulnerability Disclosure (CVD)

The practice of Coordinated Vulnerability Disclosure has evolved over three decades in the cybersecurity community, establishing robust frameworks for identifying and addressing security flaws in software systems. CVD facilitates collaboration between software vendors and vulnerability reporters to address security issues that cannot be eliminated during production (Cattell et al., 2024b). This process enables external parties and security researchers to disclose vulnerabilities through established channels, allowing organizations and the public to proactively address issues before malicious exploitation (Cybersecurity Coalition, 2019).

The significance of CVD extends beyond industry practice into government policy and regulation. In the United States, civilian government agencies and federal IT contractors are mandated to adopt vulnerability disclosure policies. The Food and Drug Administration (FDA) incorporates CVD into its guidance for medical device security (U.S. Food and Drug Administration, 2023), while key standards from the National Institute of Standards and Technology (NIST) endorse CVD as a core practice (National Institute of Standards and Technology, 2018). Internationally, the EU's NIS 2 Directive requires critical infrastructure entities to establish CVD processes (European Parliament and Council, 2022), and the EU Cyber Resilience Act mandates software manufacturers to adopt CVD processes (European Union, 2023).

Across various industries, vulnerability disclosure by third parties has demonstrably improved security (Gal-Or et al., 2024; Walshe & Simpson, 2022) and greatly accelerated corporate patch releases (Arora et al., 2010). CVD operates within a broader ecosystem of programs that identify, index, and communicate security vulnerabilities, including the National Vulnerability Database, Common Vulnerabilities and Exposures (CVE) system, and Common Vulnerability Scoring System

(CVSS) (Aggarwal, 2023). CVE Numbering Authorities are organizations authorized to issue CVEs within their scope, with MITRE acting as a "root" authority that handles disputes (Apache Software Foundation, 2024). The CERT Coordination Center at Carnegie Mellon University has been foundational in establishing these practices, with their comprehensive guide to coordinated vulnerability disclosure (Householder et al., 2017a) providing detailed frameworks for multi-party coordination that have been widely adopted across the security community.

However, the direct application of CVD to AI systems faces unique challenges – the probabilistic and complex nature of machine learning models makes judgments difficult, as models lack distinct exploited and non-exploited states (Barbierato & Gatti, 2024). Most consumer ML models are deployed as closed-source APIs (Allison, 2023), posing challenges in tracking and indexing vulnerabilities, particularly in Software as a Service (SaaS) products where versioning is not apparent to end users and issues may not be reproducible after remediation (Nichols, 2021). While MITRE recognizes some ML CVEs, the existing coordinated disclosure infrastructure lacks precendence for many ML-specific cases (Cattell et al., 2024b). Recent work extends CVD frameworks to AI systems, documenting lessons learned from coordinating ML-specific vulnerabilities (Householder et al., 2024), and recent AI red-teaming work has explicitly connected these practices to cyber red-teaming traditions (Sinha et al., 2025).

Multi-party coordination presents particular challenges when flaws affect multiple AI systems simultaneously. The Vultron protocol (Householder, 2022) provides a formal state-based model for multi-party coordinated vulnerability disclosure (MPCVD), addressing the complexity of coordinating among multiple vendors, coordinators, and affected parties (Householder & Spring, 2022). This challenge is particularly relevant to AI systems where universal vulnerabilities often impact multiple models or providers, motivating our work to improve interoperability and coordination across the fragmented ecosystem we document in Section 4.

### A.2. Bug Bounty Programs

Bug bounty programs are a critical mechanism for incentivizing users and security researchers to discover and responsibly disclose security vulnerabilities. Security vulnerability reporting has accumulated millions of volunteer researchers worldwide, thousands of organizations hosting disclosure and bug bounty programs, and millions in paid rewards annually (Longpre et al., 2024b). These programs provide financial incentives for proactive identification of flaws, demonstrably enhancing security outcomes across industries (Gal-Or et al., 2024). Key elements of effective programs include clearly defined scope documents, reward structures that incentivize discovery, and established rules of engagement that codify "good-faith research" (Oakley, 2019; Akgul et al., 2023).

Scholars have called for the adoption of bug bounties in algorithmic contexts beyond traditional software security, such as social media platforms (Eslami et al., 2019; Elazari, 2018). Early efforts in the AI domain include bias bounty programs like Twitter's algorithmic cropping bounty (Yee et al., 2021) and QueerInAI's queer bias bounty (Dennler et al., 2023). More recently, AI developers have begun implementing AI bug bounties: Google's Bug Hunter Program includes AI systems and covers privacy/security attacks and AI-specific vulnerabilities like weight extraction and prompt injections (Google, 2023), Microsoft also operates an AI bounty program (Microsoft, 2023), OpenAI runs a bug bounty administered by BugCrowd focusing on security flaws in APIs and ChatGPT (OpenAI, 2023), and Anthropic launched a model safety bug bounty program via HackerOne, though it remains invite-only (Anthropic, 2024).

However, current AI bounty programs have significant limitations. Some disclosure pathways are invite-only or exclude important AI flaws from their scope entirely, with many programs focusing primarily on traditional security vulnerabilities while excluding broader sociotechnical concerns like bias, fairness, or misinformation (Longpre et al., 2024b). Content issues and misuse are frequently out of scope for bounties, with feedback redirected through separate channels. This creates gaps in coverage for the full spectrum of AI risks that require attention. Empirical analysis of AI vendor practices reveals significant gaps in vulnerability disclosure infrastructure. (Piao et al., 2026) measured the vulnerability disclosure policies of AI vendors and found substantial variation in policy quality, scope, and accessibility. Many vendors lack clearly defined disclosure channels, while others restrict reporting to narrow categories of security vulnerabilities. This analysis highlights the need for standardized disclosure frameworks that can be consistently implemented across AI providers.

### A.3. Third-Party Evaluation and Red Teaming

As ML stakes grow, several approaches have emerged for evaluating AI systems, including cooperative audits (Wilson et al., 2021), participatory audits (Dennler et al., 2023), and AI red teaming (OpenAI, 2023; Ganguli et al., 2022). These approaches exist along a spectrum of independence from first-party (in-house) evaluation, to second-party (contracted)

evaluation, to third-party (independent) evaluation (Longpre et al., 2024b). Third-party evaluation provides benefits not available through first and second-party evaluation: broader researcher participation given the much larger set of potential evaluators outside system providers' organizations, greater coverage of evaluations given incomplete representations of perspectives and expertise within system providers, evaluator independence given the absence of conflicts of interest, and evaluation speed as external researchers can respond quickly to emerging issues (Raji et al., 2022; Reuel et al., 2025). Policy discussions on AI safety often center around pre-deployment evaluation by internal first-party evaluators or contracted second parties, but this overlooks the importance of independent third-party evaluations.

Post-deployment evaluation mechanisms are increasingly recognized as essential complements to pre-deployment testing. (Dai et al., 2025) propose aggregated individual reporting as a method for post-deployment evaluation, enabling users to report issues they encounter while using AI systems. This approach recognizes that diverse users interact with AI systems in varied contexts that cannot be fully anticipated during development, making ongoing evaluation critical for identifying emergent risks. Recent empirical work demonstrates the feasibility and value of independent third-party evaluation. (Behzad et al., 2026) conducted an external fairness evaluation of LinkedIn's Talent Search ranking system, focusing on potential bias across gender and race. Their independent audit revealed insights that complemented the platform's internal evaluations, illustrating how third-party researchers can contribute unique perspectives and findings that enhance system accountability.

Recent initiatives demonstrate the value of third-party evaluation at scale. The Generative Red Team 2 (GRT2) event at DEFCON 32 served as a comprehensive test of coordinated flaw disclosure frameworks, with participants receiving detailed model cards specifying intent and scope, preparing statistically valid reports through a dedicated platform, and having disputes mediated by an independent adjudication panel (McGregor et al., 2024b; Cattell et al., 2024b). This pilot demonstrated the feasibility of structured third-party evaluation at scale.

### A.4. Legal and Policy Frameworks: Safe Harbor and Research Protections

Safe harbor provisions shield good-faith security researchers from legal risk when reporting vulnerabilities responsibly (Tschider, 2024; HackerOne, 2023). Legal scholars and practitioners have underscored these protections as vital to fostering security research (Abdo et al., 2022). U.S. Copyright Office exemptions to DMCA Section 1201 provide limited allowances for research on protected systems (Herman & Gandy Jr, 2006; Colannino, 2021). Similarly, policy initiatives by the OECD and the EU promote researcher immunity when acting within authorized disclosure frameworks (OECD, 2024; European Council, 2024).

### A.5. AI Vulnerabilities, Incidents, and Risk Taxonomies

Efforts to systematically catalog AI vulnerabilities and incidents are emerging but remain fragmented. The AI Incident Database (AIID) (McGregor, 2021) aggregates publicly reported harms, while the AI Vulnerability Database (AVID) collects user-submitted cases across Security, Ethics, and Performance (SEP) dimensions. MITRE's ATLAS framework (Liaghati, 2025) documents adversarial tactics against ML systems, though adoption in CVE processes remains limited, with only two ML-related CVEs recorded to date (NIST, 2020; 2023).

Other global initiatives, such as the OECD AI Incidents Monitor (OECD, 2025) use monitoring to track realized incidents and latent hazards. Agencies like CISA and CERT have begun to integrate AI-related vulnerabilities into existing coordination workflows (CISA, 2022; Householder et al., 2024). CERT Coordination Center has also adopted machine-readable formats for vulnerability disclosure, with vulnerability notes now compliant with the Common Security Advisory Framework (CSAF), enabling automated processing and integration across security tools and platforms. Scholarly analyses advocate for hybrid reporting mechanisms combining incident databases, vulnerability tracking, and policy coordination (Dixon & Frase, 2024a; 2025).

Taxonomies of AI risks and harms are extensive: Weidinger et al. (Weidinger et al., 2022; 2021), Shelby et al. (Shelby et al., 2023), NIST's AI RMF (Tabassi, 2023), the MIT AI Risk Repository (Slattery et al., 2024), and Longpre et al.'s cheatsheet (Longpre et al., 2024a) are notable examples. OWASP's LLM Top 10 (OWASP, 2024) and MITRE's CWE/ATLAS resources inform technical classification too (Nie et al., 2020; Liaghati, 2025).

### A.6. Documentation, Transparency, and Governance

Model Cards and similar documentation practices support transparency and accountability (Mitchell et al., 2019; Crisan et al., 2022). Policy frameworks like the U.S. Executive Order on AI, the EU AI Act, and local laws emphasize auditability but stop

short of a unified disclosure framework comparable to CVD (Executive Office of the President, 2023; Skowron & Biderman, 2023; New York City Council, 2021). Scholars argue for interoperable reporting infrastructures linking governance, incident tracking, and technical safety (Liang et al., 2022; Seger et al., 2023).

### A.7. Limitations and Research Gaps

Existing AI flaw reporting mechanisms remain fragmented and inconsistent (Cattell et al., 2024a; Householder et al., 2024). Researchers face redundant reporting burdens across platforms, and many systems silo findings rather than coordinating cross-stakeholder disclosure (Dixon & Frase, 2024b). The reporting culture lags behind established software CVD norms, motivating the need for a unified, interoperable disclosure infrastructure bridging existing systems and clarifying norms for what counts as reportable flaws.

## B. AI Flaw & Incident Reporting System Analyses

### B.1. AI Reporting Systems

Table 3 provides an overview of the incident reporting forms currently available from various organizations that we included in our audit of existing reporting systems, including their operators and intended purposes.

*Table 3.* **AI Incident Reporting Form List**

| FORM NAME | LINK | DESCRIPTION |
| --- | --- | --- |
| **Google BugHunters AI VRP** | `https://bughunters.google.com/report/vrp` (Accessed January 20 2026) | Operated by Google, this form allows users to report security vulnerabilities related to Google's AI products and services with a bug bounty program and coordinated vulnerability disclosure. |
| **OpenAI Report Content** | `https://openai.com/form/report-content/` (Accessed January 20 2026) | Operated by OpenAI, this internal feedback form allows users to report AI-related content violations that do not align with expectations or policy. It is designed for general users and developers who interact with OpenAI's systems. |
| **OpenAI Bugcrowd Form** | `https://bugcrowd.com/engagements/openai/submissions/new` (Accessed September 23 2025) | This form is hosted through Bugcrowd, a third-party bug bounty and coordinated vulnerability disclosure platform. It enables hackers and security researchers to responsibly disclose security vulnerabilities in OpenAI's infrastructure and products. Bugcrowd provides triage and coordination for handling reports. |
| **Anthropic HackerOne Form** | `https://hackerone.com/297a385f-b3bd-4ecd-9466-7d9ad55371ce/embedded_submissions/new` (Accessed July 20 2025) | Anthropic's vulnerability disclosure form, which is managed through HackerOne: a third-party bug bounty and coordinated vulnerability disclosure platform. It supports responsible reporting by hackers and security researchers of security vulnerabilities related to Anthropic's models and services. |
| **MITRE ATLAS** | `https://ai-incidents.mitre.org/incidents/create` (Accessed July 20 2025) | The MITRE ATLAS (Adversarial Threat Landscape for Artificial-Intelligence Systems) platform, maintained by the MITRE Corporation, enables reporting of adversarial incidents in AI across vendors. It focuses primarily on adversarial ML threats, and it aims to build an understanding of real-world attack vectors and defensive mitigations. |
| **AIAAIC** | `https://www.aiaaic.org/aiaaic-repository/submit-incidentcontroversy` (Accessed July 20 2025) | The AI, Algorithmic and Automation Incidents and Controversies (AIAAIC) Repository is a publicly available database for AI-related incidents across vendors, developed by independent researchers. It catalogues AI-related incidents and harms across domains, with a goal of improving transparency and accountability. |

*Continued on next page*

Table 3 – *Continued from previous page*

| FORM NAME | LINK | DESCRIPTION |
|---|---|---|
| **CERT CC** | https://kb.cert.org/vuls/vulcoordrequest/ (Accessed July 20 2025) | The CERT Coordination Center (CERT/CC), operated by Carnegie Mellon University's Software Engineering Institute, provides a form for coordinated disclosure of security vulnerabilities. It supports researchers and vendors in responsibly coordinating and managing vulnerability information, including for AI-related software systems. |
| **CISA** | https://myservices.cisa.gov/irf?id=irf_incident_response_form (Accessed July 20 2025) | Managed by the U.S. Cybersecurity and Infrastructure Security Agency (CISA), this form enables organizations to report cybersecurity incidents, especially those pertaining to government and critical infrastructure systems. It facilitates government coordination and support for response, mitigation, and threat analysis. |
| **AVID** | https://airtable.com/app0GzVVTRKUUNUDd/shrOCPagOzxNpgV96 (Accessed July 20 2025) | The AI Vulnerability Database (AVID) is a public repository cataloging AI failures, vulnerabilities, and harms across vendors. It aims to collect and analyze incidents in accordance with structured taxonomies to promote transparency and understanding of trends. |
| **0DIN (Mozilla)** | https://0din.ai/vulnerabilities/new (Accessed August 10 2025) | 0DIN is a bug bounty and coordinated disclosure platform specific to AI systems. It enables responsible disclosure of AI-related failures and coordinates between reporters and affected vendors. |
| **OECD** | See report available at https://www.oecd.org/en/publications/towards-a-common-reporting-framework-for-ai-incidents_f326d4ac-en.html (Accessed July 20 2025) | Developed by the Organisation for Economic Co-operation and Development (OECD), this framework provides a standardized approach to AI incident reporting across countries and institutions. While not yet implemented as a public open-submission form, it is currently being used by some countries (e.g., the Netherlands) for internal information sharing. |
| **AIID** | https://incidentdatabase.ai/ (Accessed September 12 2025). | The AI Incident Database (AIID) is a public incident repository that aggregates incident reports across vendors which have appeared in news media. It serves as a central source of evidence for AI incidents and enables trend analysis through the application of structured taxonomies to incidents. |

## B.2. Dimensions of Comparison

To conduct an audit of existing reporting forms, we examine two key dimensions. Table 4 defines the information categories that may be collected by incident reporting forms. Table 5 outlines the structural characteristics of the reporting forms themselves, including scope, anonymity options, and disclosure policies. In Table 4, forms are considered to cover only explicitly requested data elements. Fields are marked as inferred when automatically collected by the receiving entity or entirely implicit in submission details. Some systems (e.g., AIID) derive elements post-submission, and users may add unrequested data in free text fields.

*Table 4.* **AI Incident Report Form Indicators**

| Form Indicator | Indicator Criteria |
|---|---|
| **Reporter Information** | Information about the individual or organization submitting the report, such as a name, email, or affiliation. |
| **System Identification** | Identification of the system where the issue was found, such as GPT-4 or Claude 3.7 Sonnet. |
| **Developers/Deployers** | The organizations or individuals who created or deployed the system. |
| **Time** | The date and time when the issue was discovered or occurred. |
| **Location** | The geographic or network location where the issue was observed. |
| **Flaw Description** | An explanation of the vulnerability, error, or unexpected behavior. |
| **Impact from AI System** | The effects or harms caused by the AI system on people, processes, or infrastructure. For example, the issue may cause financial or physical harm. |
| **Impact on AI System** | The effects on the system from the issue. For example, many forms distinguish between confidentiality, integrity, and availability impacts. |
| **Impact Severity** | The seriousness of the issue in terms of potential damage or risk. |
| **Impact Prevalence** | The extent to which the issue is widespread or frequently occurring. |
| **Stakeholders Affected** | The groups, organizations, or individuals impacted by the issue—for example, which stakeholders may experience physical or financial harm. |
| **IT Systems Affected** | Components or infrastructure beyond the impacted system itself that were impacted. |
| **Dependency Info** | Information about related third-party systems, libraries, or models involved and necessary for reproducing the issue. |
| **Discovery Method** | The way in which the issue was identified, such as testing, monitoring, or reporting. |
| **Discovery Sources** | The original channels or places where data/evidence of the flaw was first reported or noted, such as server logs or AI system outputs. |
| **Policy Violation** | Rules, regulations, or standards that the issue violates. |
| **Intended Behavior** | What the system was expected or designed to do under normal operation. |
| **Adversary Tactics** | Techniques or strategies an attacker used or could use to exploit the flaw. |
| **Weaknesses/Sources** | Underlying technical causes of the issue, such as misconfigurations, design flaws, or coding errors. |
| **Other Details** | Additional contextual or relevant information not captured elsewhere in the report. |
| **Mitigations** | Steps taken or recommended to reduce, remediate, or manage the risk. |
| **Reproduction Steps** | Instructions for reliably replicating the issue. |
| **Public Disclosure Plan** | The reporter's plans or strategies for sharing information about the flaw with the public. |
| **Relationship to System** | How the reporter or analyst is connected to the system. |
| **Flaw Evidence** | Supporting materials or proof that demonstrate the existence of the issue. |
| **Reporting Links** | References or URLs where the issue was submitted or documented, such as links to news articles about the issue. |
| **Attacker Access** | The level of access an attacker would need to exploit the flaw. |
| **Publicly Known?** | Whether the issue has already been disclosed or is publicly available. |
| **Industry/Sector** | The industry or domain in which the system or issue is situated. |

*Table 5.* **AI Incident Reporting Form Characteristics**

| Form Indicator | Indicator Criteria |
|---|---|
| **Form Exists?** | Whether a formal reporting form is available for submitting incidents or vulnerabilities. |
| **Allows Reporter Anonymity?** | Whether the reporting process permits anonymous submission without disclosing identity. |
| **Reportable Systems** | The types of systems the form accepts reports for (e.g., a specific vendor's systems or any system). |

Table 5 – *Continued from previous page*

| FORM INDICATOR | INDICATOR CRITERIA |
|---|---|
| **Scope** | The categories of issues the form is designed to capture, such as vulnerabilities, hazards, or safety/security incidents. |
| **Taxonomies** | The classification frameworks used for categorizing incidents or flaws within the report, such as harms (e.g. economic, psychological), adversary tactics (e.g. privilege escalation), or impacts on systems (e.g. availability, confidentiality). |
| **Required Fields** | The minimum number of fields that must be completed for a submission to be accepted. A range is provided if there are conditionally required fields. |
| **Total Fields** | The full number of fields included in the form, required or optional. A range is provided if there are conditional fields. |
| **Info Covered** | The quantity of information categories the form captures from Table 4. |
| **CSAM Guidance?** | Whether the form provides guidance for handling child sexual abuse material (CSAM) if encountered. |
| **Sharing & Coordination** | Whether the stakeholders maintaining the form have processes for sharing reports with other stakeholders (e.g., affected vendors, governments) and coordinated disclosure policies. |
| **Public Disclosure** | Whether the form or process includes a pathway for making reported issues public. |

## C. Our AI Flaw Reporting System

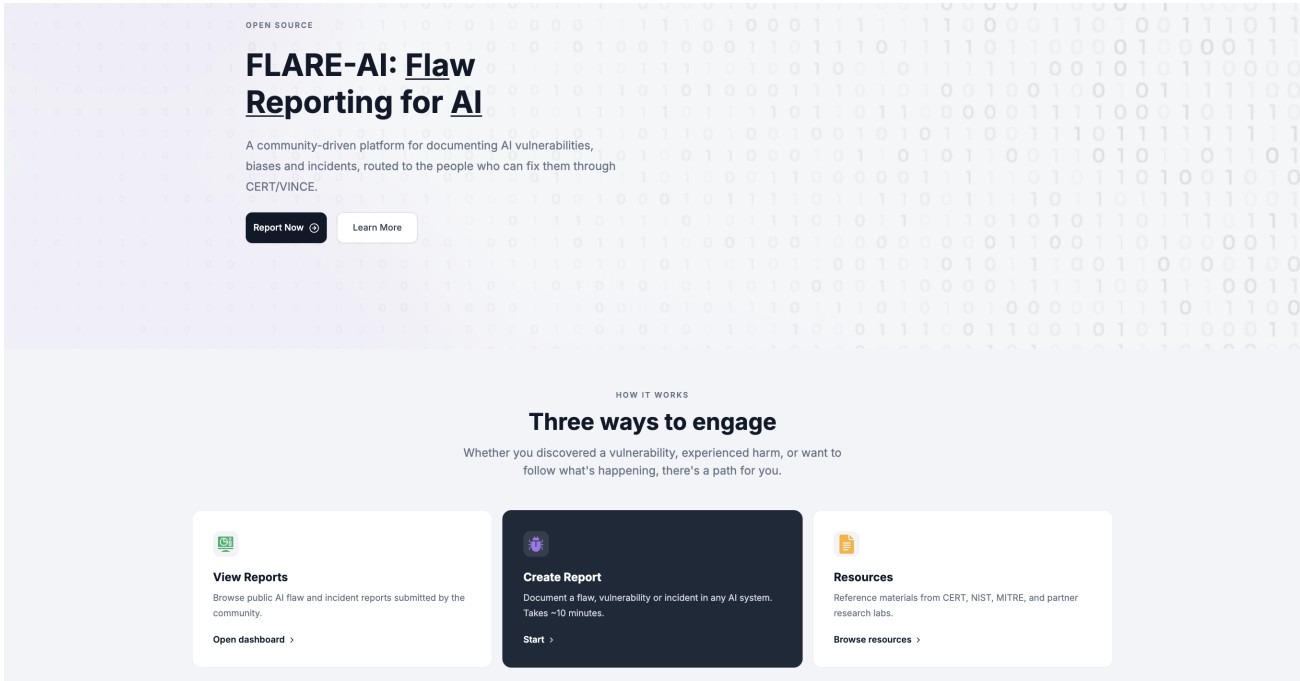

*Figure 4.* FLARE-AI home page.

The AI Flaw Reporting framework implements several key design choices that directly impact both user experience and recipient triaging capabilities. Rather than requiring users to navigate compelx technical taxonomies, the system employs a two-stage classification approach that narrows the reporting scope progressively based on key characteristics of the incident.

## C.1. Adaptive Form Logic and User Guidance

### C.1.1. CONDITIONAL BRANCHING DESIGN

The framework begins with three initial questions in Stage 1, with the first serving as a critical safety check: whether the report involves child sexual abuse material (CSAM), which triggers specialized guidance and blocks standard submission. The subsequent two classification questions determine the entire form structure.

1. Whether the flaw involves an incident where harm has already occurred.

2. Whether the flaw could be exploited by threat actors with malicious intent.

This binary classification approach serves two purposes: it reduces cognitive load on reporters while simultaneously ensuring that reports are categorized appropriately for different recipient workflows. The conditional logic reports to four report types – Real-World Incidents, Security Incident Reports, Vulnerability Reports, and Hazard Reports – each optimized for specific stakeholder needs.

### C.1.2. PROGRESSIVE DISCLOSURE AND CONTEXTUAL GUIDANCE

Rather than presenting users with an overwhelming comprehensive form, the system reveals relevant sections progressively. This design choice addresses a critical usability challenge in technical reporting: balancing comprehensiveness with accessibility. Users see only the fields relevant to their specific incident type, reducing abandonment rates while ensuring complete data collection for triaging.

The framework also implements dynamic policy mapping, automatically surfacing relevant terms of service and acceptable use policies based on the AI systems selected. This guidance helps reporters identify policy violations more accurately, improving the quality of submissions for legal and compliance teams.

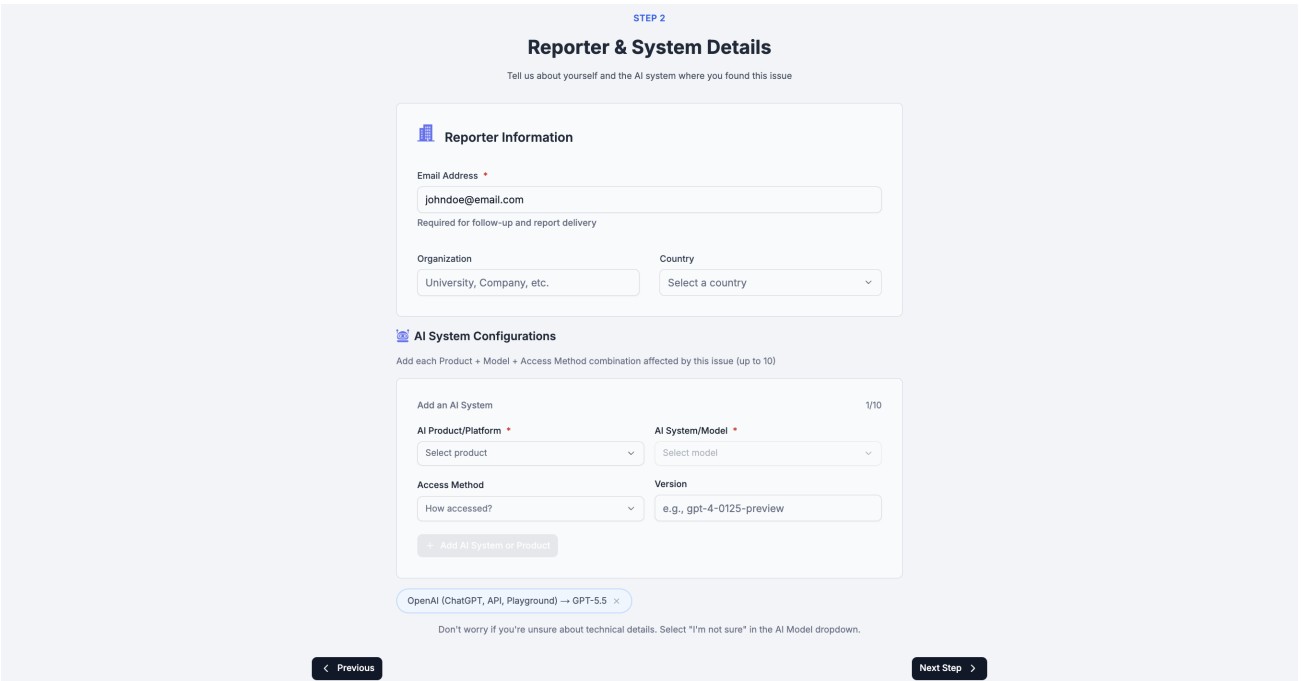

*Figure 5.* Step 2 of the reporting form, which allows users to input their contact and affected system information.

## C.2. Taxonomy Design

### C.2.1. Hierarchical Impact Classification

Our harm taxonomy, based on AIAAIC (Abercrombie et al., 2024), encompasses physical and psychological harms, as well as economic, environmental, and reputational impacts. The taxonomy distinguishes between physical and non-physical (psychological) harms to enable appropriate severity assessment and routing. Once a top level category is selected, Flare-AI dynamically presents specific subcategories. This approach addresses a key tension in flaw reporting: the need for granular classification without overwhelming non-expert users.

For example, selecting "Psychological" impacts reveal specific options like "Harassment/abuse/intimidation", each with precise definitions. This structure enables automated routing to appropriate subject matter experts while maintaining user comprehension. The machine-readable format includes semantic relationships between AI systems, harm types, and affected stakeholders, enabling automated analysis across large datasets.

### C.2.2. Cross-Platform Data Mapping

The framework implements conversion utilities that transforms reports into formats expected by different recipients. For example, reports can be automatically converted to CERT Coordination Center vulnerability report formats, among other custom incident tracking schemas. This interoperability reduces friction for recipients and increases the likelihood of appropriate action. The transformation process preserves semantics while adapting presentation to recipient-specific requirements. For example, recipients may use differing taxonomies that can be mapped onto each other. Some recipients may only be interested in a subset of fields. Incident reports often include information about both flaws with AI systems and corresponding harms, whereas vulnerability reports may only include flaw-related information.

## C.3. Sensitive Content Handling

### C.3.1. Automated Sensitive Content Detection and Routing

The system implements logic for detecting and handling reports involving illegal or highly sensitive content, particularly child sexual abuse material (CSAM). In addition to blocking these reports entirely, the system provides appropriate routing guidance to specialized authorities.

The design choice reflects a balance between comprehensive flaw coverage and responsible disclosure practices. The system recognizes that AI systems may generate harmful content, but ensures that such reports are directed to appropriate channels rather than standard developer feedback mechanisms.

## C.4. Machine-Readable Report Format and System Integration

To enable the automated dissemination and coordination described in Subsection 5.4, Flare-AI produces machine-readable reports structured as JSON and JSON-LD. This section provides technical information on the report format and integration pathways.

### C.4.1. Report Structure

Core required fields of a Flare-AI report include: reporter identifier, system identification, flaw classification, description, impact categories, and timestamp. Optional contextual fields support enhanced triage: session identifiers for log retrieval, reproduction steps, statistical validation metrics, affected stakeholder tags, geographic context, policy violation references, evidence files with descriptions, and public disclosure status.

Flare-AI reports can be downloaded in two formats. Standard JSON contains all collected information in a machine-readable structure suitable for analysis, dissemination, and integration with existing systems. JSON-LD (JSON "linked data") extends this with semantic relationships between entities, enabling richer automated reasoning and interoperability. Listing C.4.1 shows an example report in JSON-LD format.

The JSON-LD format combines vocabulary from two sources. Generic fields such as report name, description, and author use common Schema.org vocabulary, while AI-specific elements use custom Flare-AI vocabulary hosted with the application. This hybrid approach enables benefiting from common semantic web practices and domain specificity. For example, a report implicating a specific AI system (`flare:aiSystem`) can link to its developer's web presence, version information, and

other metadata via a standard Schema.org `SoftwareApplication` object, enabling automated cross-referencing across datasets. The raw JSON report is also embedded within the JSON-LD object as `flare:rawReport` for compatibility with recipients that do not parse linked data.

*Listing 1.* **Example FLARE-AI Report JSON-LD.**

```
{
  "@context": [
    {
      "schema": "https://schema.org/",
      "flare": "https://ai-reports.org/vocabulary/"
    }
  ],
  "@type": "flare:AIFlawReport",
  "@id": "https://ai-reports.org/reports/2e98e6e3-7a4c-4f5b-9d2e-8
    c1a6b3f4e7d",
  "schema:name": "AI Flaw Report: OpenAI (ChatGPT, API, Playground) - GPT
    -3.5-Turbo",
  "schema:description": "I extracted the complete embedding projection layer
     (final layer weights) from production language models by exploiting API
     features that provided logit probabilities and logit bias parameters.
    My attack recovered precise model architecture details (hidden
    dimensions) and weight matrices with mean squared error of 10^-4. I
    extracted entire projection matrices from OpenAI's ada and babbage
    models, confirming hidden dimensions of 1024 and 2048 respectively.",
  "schema:dateCreated": "2025-09-18T18:15:04.634293",
  "schema:identifier": "2e98e6e3-7a4c-4f5b-9d2e-8c1a6b3f4e7d",
  "flare:aiSystem": [
    {
      "@type": "schema:SoftwareApplication",
      "@id": "https://platform.openai.com/docs/models/gpt-3-5-turbo",
      "name": "OpenAI (ChatGPT, API, Playground)",
      "version": "GPT-3.5-Turbo",
      "description": "OpenAI (ChatGPT, API, Playground) - GPT-3.5-Turbo"
    }
  ],
  "flare:severity": "high",
  "flare:prevalence": "uncommon",
  "flare:impacts": [
    "documented"
  ],
  "flare:reportType": [
    "Malicious Use",
    "Real-World Incidents"
  ],
  "flare:policyViolation": "This attack violates the implicit expectation
    that model weights and architecture details remain proprietary. It
    potentially violates terms of service regarding reverse engineering and
    extracting confidential information about model internals. The attack
    undermines the \"black box\" nature that API providers expect to
    maintain.",
  "schema:author": {
    "@type": "schema:Person",
    "email": "reporter@email.org"
  },
  "flare:impactedStakeholders": [
    "developers",
    "organizations"
  ],
  "flare:specificHarmTypes": [
    "Privacy compromise - extraction of proprietary model information"
  ],
  "flare:classification": {
    "@type": "flare:ThreatClassification",
    "flare:realWorldHarm": true,
```

```
      "flare:maliciousUse": true,
      "flare:csamInvolved": false
    },
    "flare:securityAspect": {
      "@type": "flare:SecurityIncident",
      "flare:substrateRelationship": "independent_observer",
      "flare:attackerResources": "direct_query_access_black_box",
      "flare:attackerObjectives": "privacy_compromise",
      "flare:detectionMethod": "testing",
      "flare:discoveryNarrative": "The incident demonstrates that seemingly
      innocuous API parameters can be combined to extract proprietary model
      information, undermining the security model of \"black box\" API access.
       This creates precedent for model stealing attacks on production systems
      ."
    },
    "flare:evidence": {
      "@type": "flare:Evidence",
      "flare:stepsToReproduce": "Attack requires API access with logit bias
      and logprobs parameters. Uses mathematical techniques including SVD to
      extract hidden dimensions from logit vectors across multiple queries
      with varying bias parameters. Complete mathematical methodology and
      algorithms for dimension extraction and weight matrix recovery provided.
       Code available in supplementary materials."
    },
    "flare:disclosure": {
      "@type": "flare:DisclosurePlan",
      "flare:intent": "Yes",
      "flare:timeline": "Short-term (1-30 days)",
      "flare:channels": [
        "OpenAI",
        "General AI Flaw Database"
      ]
    },
    "flare:rawReport": {...}
}
```

## C.4.2. INTEGRATION WITH EXISTING SYSTEMS

FLARE-AI reports transform bidirectionally into formats used by established vulnerability databases:

**CVE/CWE Integration**   For traditional security vulnerabilities (prompt injection, model extraction, data poisoning), reports include CVE-compatible fields: vulnerability type, attack vector, privileges required, user interaction, and confidentiality/integrity/availability impacts. The responsible factors section maps to Common Weakness Enumeration categories (e.g., CWE-20 for input validation). Impact severity populates Common Vulnerability Scoring System metrics, enabling submission to CVE Numbering Authorities via MITRE.

**AVID Compatibility**   Reports map to the AI Vulnerability Database schema across Security, Ethics, and Performance dimensions. Our AIAAIC-based harm taxonomy aligns with AVID's categories, enabling automatic classification. System identification fields populate AVID's affected models database, and severity/prevalence/reproducibility metrics map to AVID's vulnerability scoring.

**CERT Coordination Center Integration**   For multi-party vulnerability coordination, reports are compatible with CERT's coordination framework and vulnerability note format, including CSAF-compliant machine-readable outputs. The system supports CERT's coordination metadata including disclosure timelines, affected parties, and coordination status.

**CISA Integration**   For government coordination, reports support CISA's Vulnerability Exploitability eXchange (VEX) format with status indicators (exploited, investigating, remediated). Coordination metadata includes public disclosure timelines, reporter contact preferences, and embargo requests. Industry/sector fields enable prioritization for critical infrastructure, and universal flaws trigger CERT's multi-party coordination protocols.

### C.4.3. AUTOMATED ROUTING

The machine-readable format enables intelligent routing: reports automatically include relevant developers when specific models are tagged, route to appropriate coordinators based on flaw type (CERT for security vulnerabilities, NCMEC for CSAM, incident databases for realized harms), direct to internal teams (security, privacy, safeguards, user support) based on impact categories, and route to region-specific compliance teams based on jurisdiction.

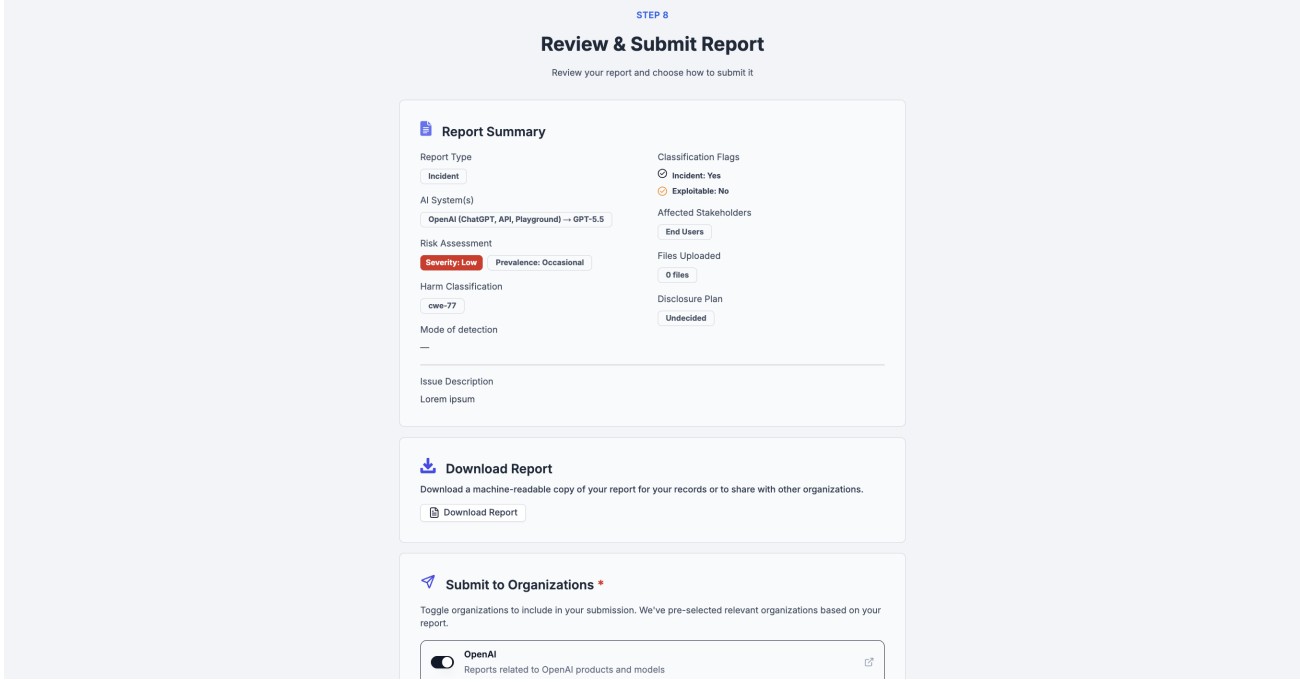

*Figure 6.* Step 8 of the reporting form, which allows users to download the report in JSON format and automatically offers options to route to different organizations based on previous selections.

### C.4.4. API INTEGRATION

Recipient organizations integrate via standardized APIs: webhook endpoints for real-time notifications, RESTful endpoints for report retrieval with filtering, status update mechanisms that propagate to reporters and coordinators, and bulk export for existing incident management systems. All APIs use OAuth 2.0 and follow OpenAPI 3.0 specifications.

Privacy protections include one-way hashed reporter identifiers, automatic PII scanning in free-text fields, granular access controls (incident databases receive public information while developers receive full technical details), and TLS 1.3 encryption for all communications.

## D. Stakeholder Consultation Process and Feedback

### D.1. Consultation Methodology

Between June and August 2025, we conducted structured consultations with 49 experts across 32 organizations (see Table 2 in the main text). Stakeholders were given demonstrations of early FLARE-AI prototypes and asked some subset of the following questions, where relevant to their expertise.

1. **Information Collection:** What set of input fields should be collected, with particular attention to what is missing or unnecessary?

2. **Intuitive Information Formatting & Taxonomies:** Whether input fields should be open text, multi-class, or multi-label selection, and what taxonomies should be used?

3. **Instructions, Ordering, & Requirements:** The provided guidance, ordering of the form information fields, and what fields should be optional vs required?

4. **Conditional Logic:** How can earlier user inputs in the form guide more efficient inferences or collection of downstream information?

5. **Report Dissemination:** How report recipients would like to receive reports: automatically via email, API, or other means?

6. **General Feedback:** In your work finding/receiving AI flaws, what other general observations, recommendations or feedback do you have for designing a new reporting system?

These questions were designed to elicit feedback to improve the ease of use of the form for *reporters*, as well as ease of triaging and coordination for *report recipients*. We often found that while reporters preferred simpler requirements, report recipients preferred more rigorous information collection.

### D.2. Representative Feedback by Stakeholder Type

**AI Developers and Report Recipients**   Organizations receiving flaw reports emphasized comprehensive information collection for effective triage. Key requests included machine-readable formats compatible with existing vulnerability databases, clear routing pathways to internal teams (security, privacy, user support, safeguards, safety), and structured taxonomies for harm categorization. Multiple developers requested session identification mechanisms beyond simple text entry, such as automatic link generation functionality. Legal compliance considerations led to requests for reporter country and jurisdiction fields, particularly for Digital Services Act obligations. Developers also emphasized the need to distinguish between issues occurring via API access, playground environments, or third-party platform integrations.

For sensitive content handling, developers stressed the importance of multiple checkpoints when users report potential CSAM, including initial warnings when relevant categories are selected and confirmation at submission that no sensitive material has been attached.

**Child Safety Organizations**   Organizations focused on child safety provided critical feedback on CSAM reporting protocols. They noted that CSAM guidelines should trigger on multiple impact categories beyond "sexualization," including "harms to children" and potentially "human rights and civil liberties," as reporters may categorize such issues in various ways. The primary concern centered on preventing inadvertent distribution of illegal material through the reporting mechanism itself.

The recommended approach involved a two-track system: directing users to submit detailed evidence (prompts, generated media) to appropriate authorities such as NCMEC or IWF with clear submission guidance, while allowing a separate sanitized report to FLARE-AI that excludes sensitive materials. Some experts suggested removing or limiting free-text entry for CSAM-related reports, restricting submissions to structured fields that indicate a CSAM risk was flagged and identify the affected model without requiring detailed descriptions that could constitute distribution of illegal content.

**Vulnerability Coordination Bodies**   Organizations specializing in coordinated vulnerability disclosure emphasized compatibility with existing security frameworks. They requested integration with or reference to CVE (Common Vulnerabilities and Exposures) and CWE (Common Weakness Enumeration) taxonomies, particularly in the "responsible factors" section of the form. This would enable seamless integration with existing vulnerability tracking systems.

Terminology refinements were highlighted as crucial. The use of "real-world" to distinguish incidents from potential flaws was identified as ambiguous and potentially confusing. Coordination bodies suggested replacing this with clearer language asking whether adverse harm has already occurred. They also recommended implementing numbered sections similar to existing CISA forms to increase familiarity for security professionals, and providing hover-over definitions for technical terms such as incident, flaw, harm, hazard, and vulnerability.

Additional recommendations included adding a field to indicate whether the flaw has already been reported to other agencies or companies and through what mechanism, enabling better coordination and avoiding duplicate efforts.

**Security Researchers and Red-Teamers**   Individuals who regularly discover and report security issues emphasized simplicity to avoid discouraging reporting. They expressed concern that overly complex forms with excessive required fields would reduce submission rates. However, they also recognized the need for comprehensive information when it could be provided efficiently.

Technical recommendations included providing "unknown" options for fields requiring specialized knowledge (such as specific AI system identification), though with user experience design that doesn't over-encourage selecting "unknown." File upload capabilities were requested with accompanying description fields for each file, rather than requiring all file context to be included in the main written description.

Security researchers noted practical issues with duplicate detection, observing that similar reports with minor variations in prompts or outputs could flood the system. They suggested implementing semantic similarity detection rather than exact matching. Some researchers expressed interest in capabilities to upload executable demonstration systems, similar to approaches used in recent AI security competitions.

Terminology consistency issues were identified, such as references to "vulnerability" in reproducibility instructions when the broader form uses "flaw" as the encompassing term.

**Academic Researchers**    Researchers contributed detailed feedback on taxonomic precision and clarity. They identified several areas of ambiguity in impact categories. The term "autonomy" was unclearwhether it referred to AI system autonomy (such as self-replication capabilities) or impacts on human autonomy. Similarly, "impacted stakeholders" included vague terms like "administrators," "vulnerable populations," and "developers" that required more precise definitions. Researchers suggested distinguishing between model developers and software developers who use AI systems.

A critical insight emerged regarding harm severity assessment. Researchers noted that harm should be evaluated at both individual and societal levels, as some issues like sycophancy or subtle misinformation might appear minor at the individual level but become severe at scale. This led to implementing separate severity assessments for individual and societal impacts.

Researchers also questioned the feasibility of certain fields. The "responsible factors" section assumes reporters can determine whether training data, feedback mechanisms, or system prompts triggered an issueinformation that may not be knowable, especially for closed systems. Suggestions included making such fields optional with clear indication that reporters should provide their best assessment.

For multi-model flaws, researchers noted confusion about whether to select each affected model individually or whether there should be a mechanism to indicate prevalence across providers. They suggested adding fields to describe desired or expected behavior alongside the observed problematic behavior.

Audience clarity was also raised as a concern. It was unclear whether the form targeted everyday users encountering unexpected AI behavior or primarily expert researchers and red-teamers conducting systematic testing. This feedback influenced the decision to design for both audiences through progressive disclosure.

**Incident Databases and Coordinators**    Organizations managing incident databases emphasized the critical importance of machine-readable formats for automated ingestion into existing systems. They expressed interest in semantic web technologies and structured data formats using schema.org types, which would enable rich metadata capture while maintaining human-readable presentation.

Cross-platform data mapping was highlighted as essential. Different recipients use varying taxonomies and schemas, requiring conversion utilities that can transform reports into formats expected by different organizations while preserving semantic meaning. Coordinators noted that incident reports often include both flaw information and harm documentation, whereas vulnerability reports may focus solely on technical flaw detailsrequiring flexible output formats.

Recommendations included clear fields for linking to existing public disclosures or detailed reports published elsewhere, enabling databases to track the full lifecycle of disclosed issues.

**Policy and Standards Organizations**    Organizations focused on policy and international standards emphasized alignment with emerging frameworks for AI incident reporting. They highlighted the importance of accommodating different legal jurisdictions and regulatory requirements across regions. Fields supporting regulatory compliance, such as jurisdiction identifiers and policy violation categorization, were seen as increasingly important as AI regulation evolves globally.

The need for interoperability with national incident reporting frameworks was stressed, as various countries develop their own AI governance mechanisms that may require compatible reporting infrastructure.

**Bug Bounty Platforms** Platform operators provided feedback on integration with existing bug bounty workflows. They emphasized that machine-readable formats are essential for automated triage and prioritization. Recommendations included examining how reports are received from existing AI-focused bug bounty programs to ensure format compatibility.

The distinction between security vulnerabilities and broader AI safety issues was highlighted as important for routingsome platforms focus exclusively on traditional security issues, while AI flaws may span security, safety, privacy, and ethical concerns requiring different expertise for evaluation.

## D.3. Key Design Tensions and Resolutions

The consultation process revealed several fundamental tensions requiring careful design tradeoffs:

- **Simplicity vs. Comprehensiveness:** Reporters wanted minimal required fields and simple submission processes, while recipients needed detailed information for effective triage and routing. This tension was particularly acute given the diverse technical expertise of potential reporters, ranging from everyday users to expert security researchers. *Resolution:* We implemented progressive disclosure with conditional logic, keeping core required fields minimal (6-8 fields) while conditionally revealing relevant additional fields based on earlier responses. Optional fields were suggested through visual highlighting rather than requirement, allowing reporters to provide additional context when available without creating barriers to submission.

- **Structured vs. Free-Text Responses:** Recipients wanted structured taxonomies for automated routing, analysis, and integration with existing systems, while reporters wanted flexibility to describe novel issues that might not fit predetermined categories. *Resolution:* We implemented hybrid approaches throughout the form, combining dropdown selections with free-text specification fields. This allows standardization for common cases while capturing unique details for novel issues. For example, impacted stakeholders include structured options plus free-text specification of who specifically is affected.

- **Expert vs. Everyday User Design:** Security professionals wanted technical precision, CVE/CWE compatibility, and detailed vulnerability information, while everyday users needed accessible language, clear guidance, and minimal technical jargon. *Resolution:* We designed the form to accommodate both audiences through multiple mechanisms: hover-over definitions for technical terms, plain language primary labels with technical terminology in parentheses, "unknown" options for fields requiring specialized knowledge, and progressive disclosure that allows experts to provide additional technical detail without overwhelming casual reporters.

- **Privacy vs. Accountability:** Allowing anonymous reporting reduces barriers to submission and protects reporters who may face retaliation, but complicates follow-up communication, verification of claims, and coordination of disclosure timelines. *Resolution:* We made contact information optional but strongly encouraged, with clear explanation of how it enables follow-up and coordinated disclosure. We added explicit privacy controls and transparency about data handling, including information about how reports might be shared with other stakeholders and under what conditions.

- **Comprehensive Scope vs. Specialized Handling:** Including all issue types (security vulnerabilities, safety hazards, privacy violations, CSAM, misinformation, etc.) in one unified form creates a single point of entry but risks inappropriate handling of sensitive content requiring specialized expertise or legal protocols. *Resolution:* We implemented conditional routing and specialized protocols that direct different issue types to appropriate channels and experts while maintaining a unified entry point. Most notably, CSAM reports trigger specific warnings, guidance to appropriate authorities, and restrictions on what can be submitted through the standard form.

- **Universal vs. Vendor-Specific Reporting:** Some stakeholders advocated for vendor-specific forms optimized for particular AI systems, while others emphasized the value of universal reporting that works across all systems and enables discovery of cross-cutting vulnerabilities. *Resolution:* We designed for universal reporting while allowing conditional logic to adapt based on selected systems. The form accommodates reports about specific vendors, multiple vendors, or unknown systems, with fields that become more or less relevant based on these selections.

## D.4. Impact on Final Design

Based on stakeholder feedback, we implemented changes across multiple design dimensions, as noted in Table 6. These refinements demonstrate the value of multi-stakeholder consultation in balancing competing priorities and ensuring the system serves diverse user needs while maintaining effectiveness for recipients. The iterative feedback process resulted

in a form that is simultaneously more accessible to everyday users and more useful to expert security researchers, while providing recipients with better structured information for triage and coordination.

*Table 6.* Summary of Major Design Changes from Stakeholder Feedback

| Initial Design | Stakeholder Concern | Final Design |
|---|---|---|
| CSAM guidelines only on "sexualization" category | May miss reports categorized under other relevant categories | Trigger on multiple categories: sexualization, harms to children, human rights |
| Single harm severity scale | Conflates individual-level and societal-level harm | Separate individual and societal severity assessments |
| "Real-world" incident terminology | Ambiguous distinction between actual and hypothetical harms | "Has adverse harm already occurred?" language |
| Impact categorization at end of form | Doesn't guide reporter's initial description | Move impact categorization before detailed description |
| No definitions for technical terms | Confusing for non-expert users | Hover-over definitions throughout form |
| Generic "stakeholders affected" | Unclear who specifically is impacted | Structured options plus free-text specification |
| Simple file upload | No context about uploaded files | File upload with accompanying description field |
| "Policy violations" label | Implies admission of wrongdoing by reporter | "Potential policy violations" framing |
| No CVE/CWE integration | Not compatible with existing vulnerability tracking systems | CVE/CWE-compatible responsible factors fields |
| Manual recipient categorization | Places burden entirely on recipients | Automated routing to appropriate teams based on form inputs |
| Required contact information | Barrier to anonymous reporting | Optional contact information with explanation of benefits |
| No system for "unknown" model | Forces incorrect selections | "Unknown" option with appropriate UX |
| Generic harm categories | Insufficient granularity for routing | Hierarchical taxonomy with broad categories and specific subcategories |
| No jurisdiction field | Cannot support regional compliance requirements | Reporter country/jurisdiction field |

