# OpenReview forum: "FLARE-AI: Flaw Reporting for AI"
_ICML.cc/2026/Conference — ICML 2026 regular_

### Official Review · Reviewer_e4Vb · 2026-03-04

**Soundness:** 3
**Presentation:** 3
**Significance:** 2
**Originality:** 2
**Overall Recommendation:** 4
**Confidence:** 4

**Summary:**

This submission studies the current ecosystem of AI flaw and incident reporting and argues that existing reporting channels are fragmented, difficult to discover, and largely non-interoperable. The authors conduct a comparative analysis of 12 prominent AI-related reporting systems spanning model developers, vulnerability coordination bodies, and incident databases, and identify five recurring design challenges: discoverability and transparency, ambiguous scope and incompatible taxonomies, inconsistent information collection, limited interoperability and coordination, and insufficient guidance for strict-liability cases. Building on this analysis and consultations with 49 experts across 32 organizations, the authors introduce FLARE-AI, an AI flaw reporting system designed to improve interoperability and streamline reporting. The system employs early-stage classification questions and conditional logic to balance reporter burden with triage-relevant detail, supports broad intake across flaws, hazards, vulnerabilities, and incidents, and generates machine-readable reports (JSON-LD) that can optionally be disseminated to multiple stakeholders from a single submission.

**Compliance With Llm Reviewing Policy:**

Affirmed.

**Final Justification:**

After considering both the paper and the authors' rebuttal, I have decided to maintain my original recommendation of a 4 (Weak Accept). The authors have addressed most of my concerns, particularly regarding the synthesis and engineering contributions. However, I still believe the paper's primary contribution is in the form of a survey and system integration, rather than new technical methods or formal frameworks. While the paper offers significant practical value, the lack of empirical evaluation limits the ability to substantiate claims regarding the system's effectiveness.

Key Points:

Synthesis and Engineering Contributions: The paper offers valuable insights into existing reporting systems and proposes a well-designed platform (FLARE-AI) based on these insights. While it does not introduce new technical methods, the system design itself is a valuable contribution grounded in expert feedback and empirical insights.
Limited Empirical Evaluation: The authors acknowledge the lack of empirical evaluation and are working to incorporate real-world feedback. While this is a step in the right direction, the current paper lacks solid evidence to demonstrate the impact of FLARE-AI in improving flaw reporting processes.
Scope of ICML: I remain unsure if this style of paper, which combines system audit and engineering design, fully aligns with ICML's typical focus on novel technical methods. If this type of work is within the scope of ICML, then I believe the paper is deserving of acceptance.

**Key Questions For Authors:**

1. Have you conducted any user studies or provided other empirical evidence to demonstrate that the proposed system indeed improves AI flaw reporting?

**Limitations:**

yes

**Strengths And Weaknesses:**

Strengths

1. Comprehensive survey and diagnostic analysis. The paper conducts a detailed comparative study of existing AI flaw reporting tools and systems, identifying key limitations and recurring design challenges. The structured analysis and synthesis of ecosystem-level issues provide useful reference value for researchers and practitioners working on AI safety and reporting infrastructure.

2. Practical system design grounded in analysis. Building on the identified challenges, the authors design and implement FLARE-AI, a unified reporting system intended to address fragmentation and interoperability issues. The system represents a concrete attempt to operationalize the survey insights and has potential practical value for improving AI flaw reporting workflows.

Weaknesses

1. The paper is largely positioned as a survey of existing systems combined with the presentation of a newly designed reporting platform. While the analysis and system design may have practical value, the contribution primarily stems from synthesis and engineering integration rather than the introduction of new technical methods or formal frameworks. Moreover, the paper does not provide empirical evaluation (e.g., user studies, reporting efficiency measurements, or triage outcome comparisons) to substantiate claims that FLARE-AI makes reporting “easier, broader, and more triage-relevant.” As a result, the impact of the proposed system is difficult to assess beyond conceptual reasoning, and it remains unclear whether the contribution aligns with the methodological novelty and technical depth typically expected at ICML.

---

> ### Author Rebuttal · Authors · 2026-03-28
>
> We would like to thank Reviewer e4Vb for their positive assessment and thoughtful feedback. We are glad they found the comparative analysis comprehensive, and insights broadly useful to both researchers and practitioners. We also appreciate the recognition that the system design is practically grounded, and that FLARE-AI “represents a concrete attempt to operationalize the insights.”
>
> **W1. Contribution Is Synthesis + Engineering Rather Than New Methods**
>
> *The contribution primarily stems from synthesis and engineering integration rather than the introduction of new technical methods.*
>
> We agree and ackowledge this work does not introduce a new technical method, but would point out many ICML papers do not (surveys, audits, evaluations, datasets, infrastructure). Our work's strength and focus is on the audit of technical safety reporting systems, assessment's by 49 expert stakeholders, infrastructure design, and construction. We think the design and construction itself is a technical contribution, grounded in empirical insights from the comparative analysis. We are happy to better emphasize the actionable elements of this work for the ML community.
>
> **W2. Limited Empirical Evaluation**
>
> *The paper does not provide empirical evaluation (e.g., user studies, reporting efficiency measurements, or triage outcome comparisons) to substantiate claims that FLARE-AI makes reporting ‘easier, broader, and more triage-relevant.’*
>
> The reviewer raises a fair point about the limited empirical evaluation, and we acknowledge this limitation. We are actively addressing it, through ongoing efforts to elicit real-world submissions from red teamers and security researchers—the initial results of which are reflected in Table 2. (Note that our existing expert assessments were framed as user studies when the experts were red teamers.) We are continuing to collect realistic submissions through these channels, and commit to incorporating additional empirical and qualitative feedback (including reporter experience and triage utility) in the camera-ready version. We believe this will substantively strengthen the evidence base for FLARE-AI’s claims around making reporting easier, broader, and more triage-relevant. However, we also hope you recognize the extensive contributions we've already made in the audits, design, and assessment of FLARE, and believe it is ready for broader academic engagement/discussion. We take the reviewer's point seriously and commit to continued work and refinement in this area, for camera ready.
>
> We appreciate your helpful and incisive feedback, and thank you for your support of our paper!

---

> > ### Author Rebuttal · Reviewer_e4Vb · 2026-04-01
> >
> > I believe the discussion of the paper has been thorough, and most of my concerns have been adequately addressed. As a systems report, the paper is excellent in its analysis of existing reporting systems and the presentation of a newly designed reporting platform.
> >
> > However, I am still uncertain whether this style of paper (a survey of existing systems combined with the presentation of a newly designed reporting platform) falls within the scope of what ICML typically accepts. I will not change my rating, as my position is simple: if this style of paper is within ICML’s acceptance scope, then the paper is deserving of acceptance.

---

### Official Review · Reviewer_xXvP · 2026-03-10

**Soundness:** 4
**Presentation:** 4
**Significance:** 4
**Originality:** 3
**Overall Recommendation:** 5
**Confidence:** 4

**Summary:**

The paper describes the challenge posed by the fragmented AI flaw reporting system we currently have and proposes the FLARE-AI solution. The solution is a centralized, open-source reporting system that is interoperable with other reporting systems and aims to address the main challenges identified by the authors through their work with multiple relevant stakeholders (such as coordination bodies, infrastructure providers, international organizations, etc.). These challenges are: the limited discoverability of the already available (but fragmented) reporting systems by the interested stakeholders; the ambiguity on what falls within and out of scope and their respective taxonomies; the inconsistency of the information that is being collected and the missing needed triage details; the limited interoperability and coordination; as well as insufficient guidance on strict-liability cases, such as child sexual abuse material. FLARE-AI's conditional logic addresses these challenges through an early classification approach and (optional) dissemination of the generated reports to the involved stakeholders. The authors acknowledge some limitations of their study (under-represented countries should be included as stakeholders, some cases might be needed to be handled on a case-by-case basis due to their nature) while also identifying some risks (centralized nature of FLARE-AI might increase attack probabilities, the system might advantage larger organizations, research liability and safe harbor protection remain when we are dealing with illegal activities). Future work and mitigation measures are also discussed.

**Compliance With Llm Reviewing Policy:**

Affirmed.

**Final Justification:**

My justification did not change.

**Key Questions For Authors:**

- The paper explains how FLARE-AI's taxonomies are designed to be extensible and interoperable, which is really helpful. However, I would appreciate a clarification on the following: to what extent can external stakeholders that already have their own reporting schemes/infrastructures adopt or integrate FLARE-AI with limited technical or organizational support? This will help me to assess FLARE-AI's interoperability at a system level.
- Could the authors clarify if/how FLARE-AI addresses formal authority reporting obligations, other than CSAM? In particular, does FLARE-AI currently support routing or formatting reports in a way that could assist with reporting to the relevant competent authority (e.g., national authorities or the AI Office), or is its role only focused on ecosystem coordination among developers/coordinators/academics? This clarification would help me assess the extent to which FLARE-AI should be understood as a coordination tool versus a compliance support tool.

**Limitations:**

yes

**Strengths And Weaknesses:**

The claims are well supported for a design and ecosystem analysis paper through comparative surveying, stakeholder feedback, and transparent discussion of limitations. The provided appendix provides a more detailed view of the general ideas and results presented in the paper's main text. The authors address each challenge they have identified in the matter with specific measures they have taken to develop FLARE-AI. At the same time, the paper doesn't demonstrate real-world deployment/testing, so we cannot assess its applicability in real-world scenarios, as noted in the Limitations and Future Work section.
The paper is clearly written, with workflow figures that explain FLARE-AI's operation and tables that compare already existing solutions with FLARE-AI. The overall narrative is easy to follow, with the authors starting by explaining the identified challenge, describing their rationale and method, listing the results of their analysis as five main challenges, and describing how they address these challenges with the design of FLARE-AI. The work is very well positioned in the context of already existing work/literature and clearly showcases how it differs from them. Additionally, the added value of FLARE-AI is also explained in-text (especially in Section 5).
The paper addresses a very prominent organizational problem that multiple AI stakeholders face. Certain incident-reporting obligations are emerging under EU regulations (e.g., the AI Act), and multiple organizations face significant uncertainty about what, when, and to whom they should report an issue. The explanation of the current landscape, identification of the main challenges, and suggestions for respective countermeasures provide a better understanding of the problem and its possible solutions. FLARE-AI itself is adaptable and can be practically implemented by other stakeholders. The scope of impact could be very broad if extended to more stakeholders, but could also be specialized, as the open-source nature of the suggested solution allows it to adapt to the specific needs of specific groups. This could serve as a solid first step towards a more centralized, standardized reporting scheme that addresses some of the challenges Europe currently faces.
The idea of reporting AI flaws on its own is not novel, but bridging the gap that currently exists in reporting AI flaws in a consistent and interoperable manner is a solid step forward in this domain. The paper offers a novel combination of existing reporting techniques and systems, integrating the best features of each with additional input from experts/involved stakeholders. This leads to a novel way of seeing reporting systems that could be further enhanced and slowly established (at least) in the (European) AI field. This new reporting system is very well articulated, with specific methodologies, challenges, and mitigation measures.

---

> ### Author Rebuttal · Authors · 2026-03-28
>
> We would like to sincerely thank Reviewer xXvP for their thorough and encouraging review. We are especially grateful for the recognition that our work addresses “a very prominent organizational problem that multiple AI stakeholders face,” and that FLARE-AI’s adaptable, open-source design could serve as “a solid first step towards a more centralized, standardized reporting scheme.” These are the goals we set out to achieve.
>
> **Q1. Adoption Effort for External Stakeholders**
>
> *To what extent can external stakeholders that already have their own reporting schemes/infrastructures adopt or integrate FLARE-AI with limited technical or organizational support?*
>
> Great question. We have purposefully designed FLARE-AI’s schema to be machine-readable (JSON-LD) for easy interoperability. This means that mapping fields and taxonomies between FLARE-AI and an existing reporting system should be straightforward, especially since we designed our form to collect a superset of triage-relevant details than existing systems (as shown in the comparative analysis in Figure 3). This lowers the barrier for organizations to integrate FLARE-AI into their existing workflows with limited additional technical or organizational support, and we are happy to make this point more explicit in the paper.
>
> **Q2. Formal Authority Reporting Obligations**
>
> *Does FLARE-AI currently support routing or formatting reports in a way that could assist with reporting to the relevant competent authority (e.g., national authorities or the AI Office)?*
>
> We do not currently format reports for specific national or regulatory bodies. However, two points are relevant. First, the form and report schema are designed to be highly extensible, and adding compliance-oriented formatting for specific authorities is a natural and feasible extension (by design). Second, among the broad set of initial recipient organizations that FLARE-AI can disseminate reports to, many already have existing commitments and infrastructure for distributing relevant reports to the appropriate authorities (certainly CERT and the major AI developers). We believe this indirect routing may in fact be preferable in the near term, as these organizations also have the triage infrastructure needed to avoid flooding national agencies with unfiltered submissions.
> We will clarify this distinction—coordination tool versus compliance support tool—more explicitly in the paper.
>
> Thank you for your support of our submission, and we look forward to addressing any further concerns!

---

> > ### Author Rebuttal · Reviewer_xXvP · 2026-04-01
> >
> > I thank the authors for their response. My questions have been fully answered, and I have nothing to add.

---

### Official Review · Reviewer_ztvD · 2026-03-20

**Soundness:** 2
**Presentation:** 2
**Significance:** 2
**Originality:** 2
**Overall Recommendation:** 2
**Confidence:** 3

**Summary:**

This paper presents Flare-AI a flaw reporting system for AI safety. It builds on the learnings from previous AI reporting systems as well as engaging relevant stakeholders from 49 experts across 32 organizations.
x

**Compliance With Llm Reviewing Policy:**

Affirmed.

**Key Questions For Authors:**

There are no major technical advancements that have been proposed in the paper. Can you please clarify the relevance of this work to this conference as compared to a conference where this work can be used as a demo and potentially have a more appropriate audience for this work?

**Limitations:**

As stated above.

**Strengths And Weaknesses:**

The paper presents a good comparison of the existing reporting systems and the novel features of the Flare AI system and the need for these features. The presentation is clear and relatively original. However, I think this paper belongs to a conference that is more aligned with an implementation/demo/tutorial track.

---

> ### Author Rebuttal · Authors · 2026-03-28
>
> We would like to thank Reviewer ztvD for acknowledging the depth of our work, its necessity, and originality. We appreciate your thoughts and engagement with the work.
>
> **Clarifying Venue Relevance:**
>
> We have thought carefully about venue fit, and would like to make the case for why this work is well-suited to ICML.
> 1. The ICML 2026 Call for Papers explicitly invites submissions on “trustworthy machine learning (reliability, fairness, safety, etc.),” and we submitted under the “Social Aspects → Accountability, Transparency, and Interpretability” track, which represents our contributions. FLARE-AI addresses a critical gap in how ML safety and security flaws are reported, triaged, and coordinated—a problem that is inseparable from the accountability infrastructure the ML community needs as these systems are deployed at scale.
> 2. We would also like to gently push back on the characterization that this work lacks technical contribution. Our relevance and technical contributions match or exceed the *many survey papers, safety evaluation benchmarks, ecosystem audits, and dataset papers* that ICML regularly and deservedly accepts. We hope to convince you our work covers multiple of these contributions.
>
> We are happy to adjust aspects of the paper to help improve the reviewer's assessment of the fit, and welcome the opportunity to address further concerns. Thank you!

---

> > ### Author Rebuttal · Reviewer_ztvD · 2026-04-03
> >
> > Thank you for the clarifications.
> >
> > While my concern still remains if ICML is a relevant platform for this paper, I see potential of this system to be used for reporting AI flaws. I will change my rating to a weak accept purely based on the system design and future potential of the work when there is sufficient feedback from the community.

---

> > > ### Author Response · Authors · 2026-04-03
> > >
> > > Dear Reviewer ztvD,
> > >
> > > Thank you for your updated assessment and for engaging thoughtfully with our rebuttal! We are glad the system design and its future potential resonated with you, and we appreciate you raising the score.
> > >
> > > We want to affirm clearly that we intend to incorporate your suggestions in the final version of the paper. While we are not able to upload a revised manuscript during the current review window, we are committed to making these changes — and we wanted to say so on the record.
> > >
> > > We also want to take this opportunity to underscore something that we feel is easy to understate in the paper as written: the design of FLARE-AI was genuinely iterative and participatory. The 49 experts across 32 organizations listed in Table 2 meaningfully shaped the system through multiple rounds of prototype demonstrations, structured feedback sessions, and design discussions over several months (March–August 2025). Partners, including several cybersecurity and AI organizations, provided direct input that changed specific design decisions (see Appendix D for details of the process and Table 6 for a summary of these changes), and several have made concrete commitments to integrate with the system. The ecosystem survey in Section 4 similarly reflects months of close analysis of 12 real reporting systems, going far beyond just literature reviews of similar reporting infrastructures.
> > >
> > > We raise this not to revisit the venue question, but because we believe it speaks to the concern your review implicitly raised: whether this is a system that people in the ecosystem would actually use. We believe the answer is yes, and that the consultation record is the strongest evidence for that.
> > >
> > > Thank you again for your time and your willingness to reconsider! We look forward to addressing your suggestions in the camera-ready version.

---

### Decision · Program_Chairs · 2026-04-30

**Decision:**

Accept (regular)

**Comment:**

The paper presents FLARE-AI, a centralized, open-source system for AI flaw and incident reporting, addressing fragmentation, inconsistent reporting practices, and limited interoperability in the AI safety ecosystem. The work is grounded in a comparative analysis of 12 existing reporting systems and iterative consultations with 49 experts across 32 organizations, resulting in a system design that operationalises practical insights into a reusable reporting infrastructure.

The paper’s primary contribution lies in systematic analysis, synthesis, and engineering integration. Reviewers note that FLARE-AI’s design is practical, clearly presented, and potentially impactful, though claims about empirical effectiveness are currently conceptual, pending broader real-world deployment and evaluation.

The design process, grounded in expert feedback and ecosystem audits, demonstrates that FLARE-AI is a feasible and thoughtfully constructed platform capable of improving reporting coordination and interoperability. Visualizations and tables in the paper clearly illustrate workflows, feature comparisons, and the reasoning behind design decisions.

Limitations include the lack of empirical evaluation of FLARE-AI’s real-world impact and the limited demonstration of adoption by external stakeholders. While these factors restrict the ability to quantify the system’s effectiveness, the authors have outlined ongoing efforts to collect submissions and feedback to strengthen empirical support.

Overall, FLARE-AI is a technically solid, well-presented, and practically valuable contribution to AI accountability. its practical value and potential for broader adoption and impact -- despite modest methodological novelty -- makes the paper a valuable contribution to ICML.